# Reason, Then Re-reason: Cross-view Revisiting Improves Spatial Reasoning

Chaofan Ma [*1]  Zhenjie Mao [*1]  Yuhuan Yang [1]  Fanqin Zeng [1]  Yue Shi [2]  Yingjie Zhou [2]  Xiaofeng Cao [3]  Jiangchao Yao [1]

## Abstract

Spatial reasoning from egocentric videos is inherently challenging because the observable evidence is constrained by the camera trajectory. Existing methods rely on single-turn inference, forcing models to resolve geometric ambiguity through semantic priors rather than verifiable evidence. We argue that spatial reasoning should be *revisitable*: conclusions formed under limited evidence should remain open to revision when complementary viewpoints become available. Building on this insight, we propose **Reason, then Re-reason** (**ReRe**), a training-free, inference-time framework with two phases: in the *Reason Phase*, an MLLM forms a spatial hypothesis from the original video; in the *Re-reason Phase*, it verifies or revises the hypothesis by observing a synthesized novel-view video. To enable effective cross-view revisiting, we design a Geometry-to-Video pipeline that renders strategically complementary novel views from predicted 3D geometry. These views feature an elevated, oblique perspective with scene-spanning coverage, while preserving the MLLM's native video interface without architectural modifications. Extensive evaluations on VSI-Bench and STI-Bench demonstrate that **ReRe** substantially boosts open-source MLLMs to rival proprietary state-of-the-art performance.

## 1. Introduction

Spatial reasoning from egocentric videos is a core capability for multimodal large language models (MLLMs). This requires models to identify objects and infer geometric constraints, relationships, and three-dimensional layouts of the space across frames and along camera motion. Yet egocentric videos provide inherently viewpoint-limited evidence. What is visible is constrained by the recorded camera trajectory, and spatiotemporal consistency may be weak or ambiguous. This makes spatial reasoning a challenging task that requires models to organize and reason over sparsely distributed evidence.

Recent works seek to improve spatial reasoning by training models with spatially grounded data or objectives (Ouyang et al., 2025; Feng et al., 2025; Liao et al., 2025). Pioneering work Video-R1 (Feng et al., 2025) employs a two-stage training paradigm with task-specific data to incentivize spatial cognition in the model. Motivated by the high cost of collecting spatial or 3D supervision, a growing line of work has explored *training-free* method to leverage MLLMs' advanced language-based reasoning capabilities. For example, See&Trek (Li et al., 2025d) improves spatial reasoning by leveraging off-the-shelf tools to extract spatial cues, organizing them into structured textual descriptions, and then relying on the MLLM for reasoning. This suggests that inference-time reasoning protocols can yield meaningful improvements even *without* fine-tuning the underlying MLLM.

Despite these advances, these methods share a common implicit assumption: spatial reasoning is performed in a *single-turn*. Given a fixed video trajectory, MLLMs must produce a final answer, and the reasoning process terminates. This assumption, however, is structurally fragile for egocentric spatial queries. As the visual evidence is strictly *trajectory-conditioned*, the temporal sequence of frames rarely aligns with the scene's actual spatial topology, leaving 3D layouts and object relations underdetermined (Yang et al., 2025a; Ravi et al., 2025). Compounding this issue, general-purpose MLLMs lack explicit 3D world modeling and only implicitly enforce cross-frame correspondence (Zheng et al., 2025a;b). When forced to answer in a single turn, models often resolve geometric uncertainty by their *semantic priors* rather than verifiable constraints. As a result, single-turn answers directly output by MLLMs are inherently *provisional* and may be *incorrect*. Ideally, such errors could be detected and corrected if complementary cross-view evidence were available (*e.g.*, a viewpoint revealing an occluded region, or disambiguating relative depth). This suggests that improving spatial reasoning is not only about better understanding in single-turn (Ouyang

---

[*]Equal contribution. Co-first authors are listed alphabetically. [1]Cooperative Medianet Innovation Center, Shanghai Jiao Tong University [2]Shanghai Jiao Tong University [3]Tongji University. Correspondence to: Jiangchao Yao <sunarker@sjtu.edu.cn>.

*Proceedings of the 43rd International Conference on Machine Learning*, Seoul, South Korea. PMLR 306, 2026. Copyright 2026 by the author(s).

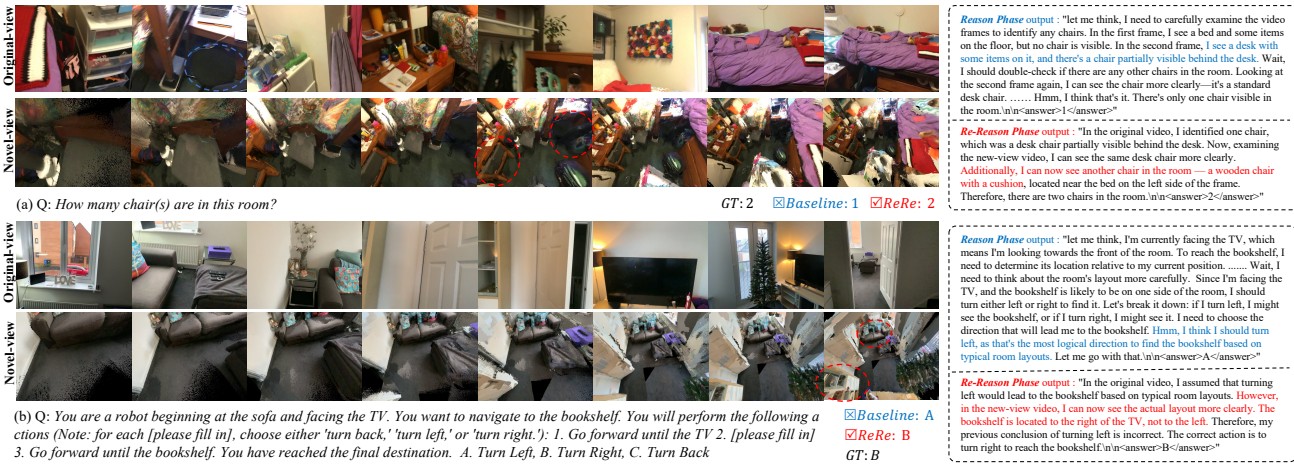

*Figure 1.* **ReRe enables the model to *revisit* its initial hypothesis under a *synthesized novel view***, correcting spatial reasoning errors that single-turn inference misses. Each case shows the original egocentric video (top frames) and the synthesized novel view (bottom frames), along with the model's reasoning before (*Reason Phase*, blue) and after (*Re-Reason Phase*, red) revisiting. *(a) Object Counting:* The synthesized elevated view reveals a chair occluded by the desk, prompting the model to correct its initial count. *(b) Route Planning:* The expanded perspective exposes a previously unobserved target, enabling the model to revise its hallucinated command.

et al., 2025; Feng et al., 2025), but about enabling the model to *revisit* its hypothesis under complementary cross-view evidence, if such evidence can be obtained cheaply at scale.

Fortunately, recent advances in monocular geometry prediction (*e.g.*, VGGT (Wang et al., 2025b)) make it feasible to recover 3D structure and synthesize novel viewpoints *at scale*. Several methods leverage such geometry to enhance spatial reasoning, typically by injecting geometric features through dedicated encoders (Wu et al., 2025; Zheng et al., 2025a; Guo et al., 2025), or with auxiliary supervision (Huang et al., 2025; Li et al., 2025a). While effective, these approaches use geometry only *implicitly*, which has two limitations. First, aligning latent geometric representations requires architectural modifications and task-specific heavy training, limiting scalability. More fundamentally, these methods remain bound to a single-turn paradigm: while geometry enriches internal representations, it does not serve as observable visual evidence that enables the model to explicitly revisit its reasoning.

In this work, we argue that spatial reasoning should be treated as a *revisitable* process: instead of committing to a single answer, the model should formulate a hypothesis and then verify it against complementary cross-view evidence. Building on this insight, we propose **Reason, then Re-reason** (**ReRe**), an inference-time framework that decomposes spatial reasoning into two phases. In the first ***Reason Phase***, the MLLM analyzes the original egocentric video to form an initial hypothesis, consisting of a thinking trace that articulates spatial observations and a provisional answer. In the second ***Re-reason Phase***, the model observes a VGGT-synthesized novel-view and explicitly validates its prior reasoning against this new evidence, retaining or

revising its former conclusion accordingly.

Notably, realizing this effective cross-view verification poses two challenges. First, the viewpoint must be strategically chosen: a randomly selected view may simply introduce different occlusions without resolving the original ambiguities. Second, the output must be directly consumable by the MLLM: while recent advances enable 3D geometry prediction, the raw point cloud cannot be natively processed by video-based models. We therefore design a ***Geometry-to-Video*** pipeline: *Trajectory Planning* synthesizes strategically complementary viewpoints, while *View Rendering* converts the predicted geometry into standard video frames. The entire protocol operates at inference time with the MLLM frozen, requiring no architectural modifications or additional training.

We validate our framework on multiple spatial reasoning benchmarks. Results show that **ReRe** yields significant performance gains across diverse open-source architectures. As illustrated in Figure 1, **ReRe** can correct spatial errors that single-turn inference cannot resolve. Further ablation studies confirm that the revisiting protocol is the primary driver of performance, and that synergizing egocentric semantics with allocentric structural evidence is indispensable for effective verification.

Our contributions are as follows:

- We identify the structural fragility of single-turn spatial reasoning from egocentric videos, and advocate for a *revisitable* paradigm.

- We propose **ReRe**, a training-free framework that structures inference into a revisiting protocol: *Reason Phase* for forming an initial hypothesis from the original video,

then *Re-reason Phase* for verifying it against synthesized cross-view evidence.

• We design a *Geometry-to-Video* pipeline that renders strategically complementary novel views to make 3D geometric cues natively consumable by frozen MLLMs.

• Extensive experiments on VSI-Bench and STI-Bench demonstrate broad improvements across diverse architectures, with ablation studies validating the necessity of cross-view synergy.

## 2. Related Work

**Visual Spatial Understanding.** Earlier research on visual spatial understanding has primarily focused on static images, aiming to ground objects and reason about their spatial layout and structure within a single frame (Kazemzadeh et al., 2014; Liu et al., 2023; Kamath et al., 2023; Ma et al., 2025a; 2022; 2023; Mao et al., 2026; Yang et al., 2024b;a; 2026). While recent MLLMs perform well on such tasks (Ma et al., 2025b; Ogezi & Shi, 2025; Chen et al., 2024a), relying solely on static imagery fundamentally limits spatial reasoning to a fixed, partially occluded viewpoint. More recently, the field has witnessed a shift towards video-based visual spatial understanding, where video serves as a sequence of spatially sampled observations that uncover the underlying 3D geometry (Yang et al., 2025a). Unlike general video tasks that focus on temporal event progression (Xiao et al., 2021; Ju et al., 2023; Wang et al., 2025a; Yang et al., 2025b), spatial video understanding treats the footage as a continuous trajectory of viewpoints, where the core challenge lies in aggregating spatial cues across frames to form a coherent environmental representation. To address this, recent efforts have diverged into two streams: training-based methods (Feng et al., 2025; Ouyang et al., 2025; Wu et al., 2025; Zhang et al., 2025) inject spatial awareness via fine-tuning on 3D-grounded data, while training-free approaches (Li et al., 2025d; Taguchi et al., 2025) convert sequential spatial cues into formats directly interpretable by MLLMs, thereby eliciting their intrinsic spatial capabilities. Despite this progress, existing approaches remain predominantly confined to a single-turn inference paradigm. They are forced to resolve geometric ambiguities from a fixed, pre-recorded trajectory, lacking the mechanism to verify their spatial hypotheses against complementary visual evidence. In contrast, our approach introduces a revisitable reasoning paradigm, actively synthesizing novel views to resolve spatial ambiguities beyond the fixed trajectory.

**Leveraging 3D Geometry for Spatial Reasoning.** Recent breakthroughs in monocular geometry prediction, particularly VGGT (Wang et al., 2025b), have demonstrated the feasibility of recovering high-fidelity 3D structures from 2D inputs at scale (Kerbl et al., 2023; Shi et al., 2021; 2025;

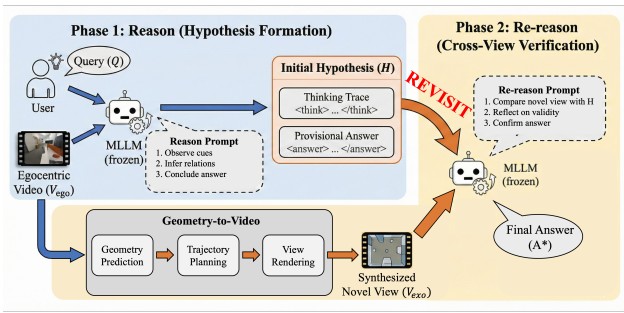

*Figure 2.* **Overview of the ReRe Framework.** Given an egocentric video, our method operates in two phases: (1) **Reason Phase**, where the MLLM forms an initial hypothesis from the original view; and (2) **Re-reason Phase**, where the model verifies its hypothesis against a synthesized allocentric view ($V_{exo}$). The *Geometry-to-Video* pipeline generates $V_{exo}$ via trajectory planning and view rendering to provide complementary geometric evidence.

Li et al., 2025c). Building upon this foundation, a growing body of work seeks to harness VGGT's geometric representations to enhance spatial reasoning. Prevalent strategies typically involve employing the VGGT encoder to extract latent spatial features, which are subsequently aligned with the MLLM's feature space (Wu et al., 2025; Zheng et al., 2025a; Guo et al., 2025), or applying auxiliary supervision from VGGT during training (Huang et al., 2025; Li et al., 2025a). While effective, these approaches utilize VGGT's capabilities only implicitly, presenting two limitations. First, aligning latent geometric representations often demands architectural modifications and costly fine-tuning. Second, and more critically, these methods remain bound to a single-turn paradigm: by treating geometry as latent context rather than explicit visual evidence, they lack the mechanism to verify spatial hypotheses against novel views, leaving them vulnerable to occlusion-induced hallucinations. In contrast, we utilize generative geometry as observable visual evidence, empowering MLLMs to explicitly verify their reasoning in a training-free manner.

## 3. Methodology

We introduce **Reason, then Re-reason** (**ReRe**), an inference-time framework designed for spatial reasoning from egocentric videos. Our core insight is that spatial reasoning should not be a single-turn perceptual task, but a *revisitable process* of hypothesis formation and verification.

### 3.1. Problem Formulation

**Task Definition.** Consider a spatial reasoning task where an MLLM $\mathcal{M}$ is given an egocentric video $V_{ego}$ and a natural language query $Q$. The goal is to predict an answer $A$ to the query (*e.g.*, object location, spatial relation, room layout). Unlike general video understanding tasks that often rely on global semantic recognition, egocentric spatial reasoning requires reasoning about 3D spatial relationships from par-

tially observable, trajectory-conditioned viewpoints.

**Standard Formulation.** Conventional approaches model this as single-turn conditional inference:

$$A^* = \arg\max_A P_\mathcal{M}(A \mid V_{ego}, Q). \tag{1}$$

This formulation assumes that $V_{ego}$ provides sufficient evidence to determine $A$. However, egocentric videos are inherently *trajectory-conditioned*: the observable evidence is constrained by the camera path, leaving the underlying 3D scene geometry underdetermined. When visual evidence is insufficient, the model must implicitly rely on learned priors to resolve ambiguity, which can lead to plausible but incorrect answers.

**Our Formulation: Revisitable Reasoning.** We reformulate spatial reasoning as a *two-phase* process that incorporates cross-view verification. Let $V_{exo}$ denote a synthesized video from a novel viewpoint that provides complementary visual evidence. We introduce an intermediate hypothesis $H$ and decompose the reasoning as:

$$
\begin{aligned}
H &\sim P_\mathcal{M}(H \mid V_{ego}, Q), \\
A^* &= \arg\max_A P_\mathcal{M}(A \mid H, V_{exo}, Q).
\end{aligned}
\tag{2}
$$

This two-phase formulation enables the model to *revisit* its initial beliefs under complementary geometric evidence before committing to a final answer. We next provide an overview of the framework and then detail each phase.

### 3.2. The ReRe Framework

Given an egocentric video $V_{ego}$ and a spatial query $Q$, the **ReRe** framework operates in two distinct phases (Figure 2): **Reason Phase** for hypothesis formation followed by **Re-reason Phase** for cross-view verification.

The first **Reason Phase** is aimed at hypothesis formation. The MLLM analyzes $V_{ego}$ and produces an initial hypothesis $H$, which contains a thinking trace $T$ and a provisional answer $\tilde{A}$. Due to the inherent viewpoint limitations of egocentric footage, this output is treated as *provisional* rather than a definitive conclusion.

The second **Re-reason Phase** focuses on cross-view verification. We first synthesize a novel-view video $V_{exo}$ from the 3D geometry recovered from $V_{ego}$, offering an allocentric perspective of the same scene. The MLLM then receives $V_{exo}$ together with its prior hypothesis $H$, and is prompted to explicitly compare this new visual evidence against its initial reasoning. Based on whether its spatial claims hold under the new viewpoint, the model confirms or revises its initial hypothesis to produce the final answer $A^*$.

Crucially, this entire protocol requires *no fine-tuning*. It purely leverages the MLLM's in-context reasoning capabilities for self-correction. We detail each phase below.

### 3.3. Reason Phase: Initial Hypothesis Formation

The goal of this phase is to obtain an initial spatial hypothesis. Given the egocentric video $V_{ego}$ and query $Q$, we prompt the MLLM $\mathcal{M}$ to produce the hypothesis $H$:

$$H = \mathcal{M}(\text{prompt}_{\text{reason}}, V_{ego}, Q), \tag{3}$$

where $\text{prompt}_{\text{reason}}$ is a task prompt that instructs the model to reason step-by-step before answering. Here, the hypothesis $H = (T, \tilde{A})$. $T$ is a thinking trace that records the model's observations and spatial inferences, and $\tilde{A}$ is a provisional answer in the task-required format (*e.g.*, a letter for multiple choice, a number for regression). Rather than requesting only a final answer, we ask the model to articulate its reasoning explicitly, making assumptions visible for later verification. We next describe the reasoning protocol that guides this process and the structured output format that captures the result.

#### 3.3.1. REASONING PROTOCOL

The key design principle is to *separate perception from reasoning* and make the model's inference process explicit and traceable. Rather than letting the model jump directly to an answer, we guide it through a structured chain-of-thought that grounds conclusions in visual evidence. Specifically, the $\text{prompt}_{\text{reason}}$ decomposes spatial reasoning into three sequential objectives: (1) *Observe*: identify and describe key visual elements, including objects, spatial arrangements, and geometric cues. (2) *Infer*: reason about plausible spatial relations based on observations, even when visual information is incomplete. (3) *Conclude*: formulate a tentative answer, explicitly framed as provisional. This decomposition ensures that the model's assumptions are surfaced in the thinking trace, providing a concrete basis for verification in Re-reason Phase.

#### 3.3.2. STRUCTURED OUTPUT

The hypothesis $H = (T, \tilde{A})$ is captured in a structured format with two tagged components: the thinking trace $T$ is enclosed in `<think>...</think>`, containing observations and spatial inferences. The provisional answer $\tilde{A}$ is enclosed in `<answer>...</answer>`. This separation serves two purposes: (1) explicit articulation surfaces implicit assumptions, often stemming from semantic priors, that are prime candidates for verification; (2) the thinking trace provides a concrete basis for the Re-reason Phase to check whether specific spatial claims hold under the new viewpoint.

### 3.4. Re-reason Phase: Cross-View Verification

With the initial hypothesis $H$ in hand, the Re-reason Phase *revisits* it under a complementary viewpoint to verify or refine the conclusion. Given the synthesized allocentric

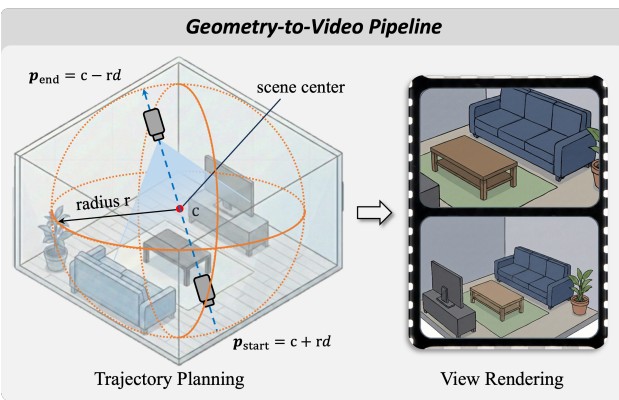

**Figure 3. Overview of the Geometry-to-Video Pipeline.** It consists of two stages: (1) *Trajectory Planning*, where we predict a 3D point cloud via VGGT and design a scene-spanning *Oblique Sweep* path; and (2) *View Rendering*, where we synthesize temporally coherent video frames $V_{exo}$ via point-based rasterization.

video $V_{exo}$, the original query $Q$, and the prior hypothesis $H = (T, \tilde{A})$, the model produces the final answer $A^*$:

$$A^* = \mathcal{M}(\text{prompt}_{\text{re-reason}}, H, V_{exo}, Q), \qquad (4)$$

where $\text{prompt}_{\text{re-reason}}$ instructs the model to explicitly compare the new evidence against its prior reasoning and revise if necessary. We next describe the re-reasoning protocol and then detail how the cross-view video $V_{exo}$ is generated.

### 3.4.1. RE-REASONING PROTOCOL

The key design principle is to enable *explicit self-correction*: the model must confront its prior reasoning with new visual evidence before committing to a final answer. Specifically, the $\text{prompt}_{\text{re-reason}}$ guides the model through three objectives: (1) *Compare*: examine the novel-view video and identify any discrepancies with the original egocentric observations. (2) *Reflect*: assess whether the spatial claims in the thinking trace $T$ still hold under the new viewpoint. (3) *Confirm*: confirm the final answer $A^*$ after determining whether to retain or revise the initial prediction $\tilde{A}$. This forces the model to ground its final decision in cross-view evidence, effectively mitigating hallucinations caused by Reason Phase.

### 3.4.2. CROSS-VIEW VIDEO GENERATION

To enable effective verification, the synthesized video $V_{exo}$ must provide geometric evidence that *complements* the original egocentric observation. Below, we first motivate our design choices, then describe our *Geometry-to-Video* pipeline, illustrated in Figure 3, which consists of *Trajectory Planning* and *View Rendering*.

**Design Principles.** To serve as effective verification, the synthesized evidence must satisfy two key design principles: (1) **Geometric Complementarity**. Since egocentric videos are trajectory-conditioned, the new view must be

strategically chosen to expose hidden spatial information. This requires a viewpoint that *reduces inter-object occlusion* and *maximizes spatial coverage*. (2) **Native Compatibility**. To leverage the pre-trained reasoning of MLLMs without architectural modifications, the geometric evidence must be presented in a *familiar visual format* rather than raw 3D representations like point clouds.

Guided by these principles, we implement a *Geometry-to-Video* pipeline with two stages: *Trajectory Planning* to ensure geometric complementarity, and *View Rendering* to ensure native compatibility. We describe each below.

**Trajectory Planning.** The goal is to design a camera path that achieves geometric complementarity, reducing occlusion and maximizing coverage. Such a path fuses scene elements dispersed across many frames of $V_{ego}$ into a single MLLM-digestible spatial summary, much like an overhead map for a maze. Imagine an aircraft flying diagonally across a city from one corner to the opposite corner, looking down at an oblique angle. This simple trajectory naturally satisfies both requirements: the *elevated* viewing angle reduces inter-object occlusion typical of eye-level egocentric recordings, while the *diagonal* path spans the full spatial extent to maximize coverage.

Concretely, we first use VGGT (Wang et al., 2025b) to predict a 3D point cloud $P_{3D}$ from $V_{ego}$. From this point cloud, we compute the scene center $\mathbf{c}$ and horizontal radius $r$ (the 95th percentile of point distances to $\mathbf{c}$ on the ground plane). The camera position $\mathbf{p}(t)$ along the path is then defined as:

$$\mathbf{p}(t) = \mathbf{c} + r \cdot (1 - 2t) \cdot \mathbf{d}, \quad t \in [0, 1], \qquad (5)$$

where $\mathbf{d} = \text{normalize}([1, \sqrt{2}, 1]^\top)$ is the diagonal direction with a $45°$ elevation angle above the ground plane. At $t = 0$, the camera starts at $\mathbf{c} + r\mathbf{d}$, and at $t = 1$, it ends at $\mathbf{c} - r\mathbf{d}$. This produces a scene-spanning, long-baseline sweep that covers the full spatial extent of the environment. The camera maintains a fixed viewing direction (aligned with $\mathbf{d}$) throughout the trajectory, ensuring stable, temporally coherent motion without disorienting rotations. We refer to this camera path design as the *Oblique Sweep* trajectory.

**View Rendering.** Since MLLMs cannot directly process point clouds, we render the predicted geometry as standard video frames, ensuring direct compatibility without architectural modifications. The main challenge is that sparse geometry, unreliable predictions, and occlusion errors can produce visual artifacts that distract or mislead the model. We address this with a simple rasterization pipeline combining three techniques: (1) *Z-buffer depth ordering* ensures correct visibility, so closer points properly occlude farther ones, preserving accurate spatial relationships. (2) *Confidence filtering* removes points with low normalized predic-

*Table 1.* **Performance on VSI-BENCH. ReRe** boosts spatial reasoning across diverse MLLMs. [†] denotes VSI-BENCH (tiny) subset.

| Methods | Avg. | Obj. Count | Abs. Dist. | Obj. Size | Room Size | Rel. Dist. | Rel. Dir. | Route Plan | Appr. Order |
|---|---|---|---|---|---|---|---|---|---|
| | | **Numerical Answer** | | | | **Multiple-Choice Answer** | | | |
| *Baseline* | | | | | | | | | |
| CHANCE LEVEL (RANDOM) | - | - | - | - | - | 25.0 | 36.1 | 28.3 | 25.0 |
| CHANCE LEVEL (FREQUENCY) | 34.0 | 62.1 | 32.0 | 29.9 | 33.1 | 25.1 | 47.9 | 28.4 | 25.2 |
| *VSI-Bench (tiny) Perf.* | | | | | | | | | |
| [†]HUMAN LEVEL | 79.2 | 94.3 | 47.0 | 60.4 | 45.9 | 94.7 | 95.8 | 95.8 | 100.0 |
| [†]GEMINI-1.5 FLASH | 45.7 | 50.8 | 33.6 | 56.5 | 45.2 | 48.0 | 39.8 | 32.7 | 59.2 |
| [†]GEMINI-1.5 PRO | 48.8 | 49.6 | 28.8 | 58.6 | 49.4 | 46.0 | 48.1 | 42.0 | 68.0 |
| [†]GEMINI-2.0 FLASH | 45.4 | 52.4 | 30.6 | 66.7 | 31.8 | 56.0 | 46.3 | 24.5 | 55.1 |
| *Proprietary Models (API)* | | | | | | | | | |
| GEMINI-1.5 FLASH | 42.1 | 49.8 | 30.8 | 53.5 | 54.4 | 37.7 | 41.0 | 31.5 | 37.8 |
| GEMINI-1.5 PRO | 45.4 | 56.2 | 30.9 | 64.1 | 43.6 | 51.3 | 46.3 | 36.0 | 34.6 |
| GPT-4O | 34.0 | 46.2 | 5.3 | 43.8 | 38.2 | 37.0 | 41.3 | 31.5 | 28.5 |
| *Open-source Models* | | | | | | | | | |
| QWEN2.5-VL-3B | 26.4 | 15.8 | 23.9 | 33.4 | 27.8 | 20.5 | 34.4 | 29.9 | 21.5 |
| **OURS** | 28.2$_{+1.8\%}$ | 16.7$_{+0.9\%}$ | 25.0$_{+1.1\%}$ | 35.3$_{+1.9\%}$ | 25.3$_{-2.5\%}$ | 31.4$_{+9.9\%}$ | 35.7$_{+1.3\%}$ | 28.9$_{-1.0\%}$ | 17.3$_{-4.2\%}$ |
| QWEN2.5-VL-7B | 24.8 | 11.2 | 12.2 | 22.8 | 29.9 | 33.8 | 36.0 | 32.0 | 24.9 |
| **OURS** | 29.5$_{+4.7\%}$ | 18.0$_{+6.8\%}$ | 18.8$_{+6.6\%}$ | 40.0$_{+17.2\%}$ | 31.1$_{+1.2\%}$ | 34.1$_{+0.3\%}$ | 35.3$_{-0.7\%}$ | 32.5$_{+0.5\%}$ | 22.0$_{-2.9\%}$ |
| QWEN3-VL-2B | 22.5 | 14.2 | 14.7 | 29.8 | 10.8 | 19.9 | 34.1 | 23.7 | 19.4 |
| **OURS** | 31.0$_{+8.5\%}$ | 17.4$_{+3.2\%}$ | 23.4$_{+8.7\%}$ | 50.5$_{+20.7\%}$ | 21.0$_{+10.2\%}$ | 25.8$_{+5.9\%}$ | 36.7$_{+2.6\%}$ | 27.8$_{+4.1\%}$ | 26.2$_{+6.8\%}$ |
| QWEN3-VL-4B | 30.7 | 14.8 | 22.0 | 41.6 | 33.5 | 30.1 | 38.4 | 23.7 | 29.5 |
| **OURS** | 36.5$_{+5.8\%}$ | 21.1$_{+6.3\%}$ | 26.7$_{+4.7\%}$ | 50.2$_{+8.6\%}$ | 35.8$_{+2.3\%}$ | 37.2$_{+7.0\%}$ | 43.1$_{+4.7\%}$ | 28.9$_{+5.2\%}$ | 34.1$_{+4.7\%}$ |
| QWEN3-VL-8B | 30.5 | 16.4 | 20.7 | 43.0 | 28.0 | 36.8 | 35.1 | 22.7 | 26.9 |
| **OURS** | 35.8$_{+5.2\%}$ | 19.5$_{+3.1\%}$ | 25.8$_{+5.1\%}$ | 49.8$_{+6.7\%}$ | 31.3$_{+3.3\%}$ | 39.7$_{+3.0\%}$ | 41.0$_{+5.9\%}$ | 29.4$_{+6.7\%}$ | 33.8$_{+7.0\%}$ |
| INTERNVL2.5-4B | 31.3 | 37.6 | 24.5 | 37.4 | 22.7 | 32.0 | 33.2 | 27.8 | 26.9 |
| **OURS** | 32.3$_{+1.0\%}$ | 35.8$_{-1.9\%}$ | 23.3$_{-1.2\%}$ | 36.9$_{-0.5\%}$ | 23.7$_{+1.1\%}$ | 32.5$_{+0.6\%}$ | 42.1$_{+8.9\%}$ | 30.9$_{+3.1\%}$ | 22.8$_{-4.0\%}$ |
| INTERNVL2.5-8B | 35.5 | 22.5 | 28.4 | 45.6 | 35.3 | 35.6 | 43.3 | 32.5 | 29.9 |
| **OURS** | 36.7$_{+1.2\%}$ | 19.1$_{-3.4\%}$ | 29.7$_{+1.3\%}$ | 46.7$_{+1.1\%}$ | 37.9$_{+2.6\%}$ | 36.5$_{+0.9\%}$ | 46.9$_{+3.6\%}$ | 31.4$_{-1.1\%}$ | 32.0$_{+2.1\%}$ |
| INTERNVL3-2B | 26.5 | 42.2 | 22.8 | 26.4 | 17.6 | 25.2 | 34.3 | 25.8 | 10.8 |
| **OURS** | 29.9$_{+3.4\%}$ | 37.9$_{-4.3\%}$ | 24.3$_{+1.5\%}$ | 27.2$_{+0.8\%}$ | 16.6$_{-1.0\%}$ | 32.0$_{+6.8\%}$ | 43.2$_{+8.9\%}$ | 26.8$_{+1.0\%}$ | 18.3$_{+7.5\%}$ |
| INTERNVL3-8B | 32.6 | 40.4 | 23.3 | 43.4 | 30.1 | 34.9 | 33.4 | 30.9 | 19.3 |
| **OURS** | 35.5$_{+2.9\%}$ | 38.8$_{-1.6\%}$ | 31.1$_{+7.8\%}$ | 44.4$_{+1.0\%}$ | 30.0$_{-0.1\%}$ | 37.2$_{+2.3\%}$ | 37.3$_{+3.9\%}$ | 30.9$_{+0.0\%}$ | 23.6$_{+4.4\%}$ |

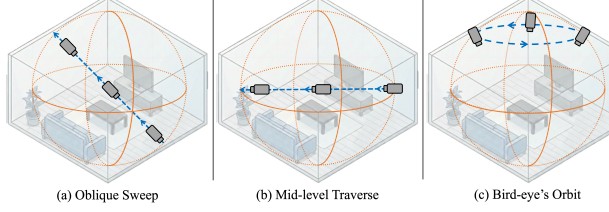

(a) Oblique Sweep    (b) Mid-level Traverse    (c) Bird-eye's Orbit

*Figure 4.* **Visual Comparison of Allocentric Trajectory Designs.** (a) *Oblique Sweep* (Ours) follows a diagonal path through the scene center with an elevated tilt. (b) *Mid-level Traverse* moves horizontally along the diameter at a fixed elevation. (c) *Bird's-eye Orbit* circles the scene center from a top-down perspective.

tion confidence (below 0.5) to suppress noise from unreliable geometry estimates. (3) *Per-frame median filtering* reduces residual salt-and-pepper artifacts.

The resulting video $V_{exo}$ presents a temporally coherent view of the scene geometry, directly compatible with the MLLM's native video interface without requiring architectural modifications or additional encoders.

**Alternative Trajectories.** While our framework is flexible regarding view generation strategies, we adopt the *Oblique Sweep* trajectory to maximize verification effectiveness. We consider two natural alternatives for constructing $V_{exo}$, as visualized in Figure 4: (1) *Mid-level Traverse*: The cam-

era translates horizontally along the diameter of the scene's equatorial plane, passing directly through the geometric center. Unlike the diagonal Oblique Sweep, this trajectory restricts movement to a single horizontal plane, close to an eye-level perspective typical of human navigation. (2) *Bird's-eye Orbit*: The camera circles the scene center at a fixed high elevation while maintaining a strictly downward-looking orientation. This trajectory provides a continuous top-down rotation, visually resembling a rotating 2D floor plan. We compare these trajectories empirically in Sec. 4.3.

### 3.5. Why *"Re-Reason"* instead of *"Joint Reason"*

Given complementary views $V_{ego}$ and $V_{exo}$, straightforward baselines involve concatenating or interleaving them for single-turn inference. However, we argue that such *passive aggregation* is suboptimal, particularly in our *training-free* setting. Without explicit fine-tuning on multi-video datasets, joint inputs represent an *out-of-distribution* modality for standard MLLMs. Consequently, unstructured data expansion, as noted by Peng et al. (2025), leads to *cognitive overload* and *conflict confusion*. When presented with contradictory visual signals (*e.g.*, varying occlusion states), even with interleaved inputs, the frozen model lacks a learned mechanism to prioritize evidence or establish robust cross-video correspondence, often defaulting to the most salient features rather than explicitly resolving the geometric ambiguity.

Specifically for concatenation strategies, splicing trajectories with distinct spatiotemporal characteristics disrupts the intrinsic *temporal coherence* of the video input. Since $V_{ego}$ and $V_{exo}$ follow fundamentally different motion logics, forcibly splicing them creates unnatural discontinuities that confuse the model's internal representation, hindering its ability to interpret the sequence as a coherent physical event.

In contrast, our sequential revisiting protocol enforces a form of *dialectical reasoning* tailored for inference-time verification. By first articulating a provisional hypothesis, the model transforms the second phase from a generic perception task into a focused *critique session*. The synthesized view $V_{exo}$ thus serves not just as "more data", but as targeted *counter-evidence* that forces the model to explicitly resolve conflicts. This *hypothesis-driven* interaction effectively structures the reasoning process, enabling robust error correction without the need for parameter updates. We empirically validate these design choices in Sec. 4.3, demonstrating that our revisiting protocol significantly outperforms joint-inference baselines.

## 4. Experiments

### 4.1. Evaluation Setup

**Datasets. VSI-Bench** (Yang et al., 2025a) constitutes a challenging benchmark designed to evaluate fine-grained spatial understanding and multi-object correspondence in video. The dataset comprises over 5,000 question-answer pairs derived from 288 real-world egocentric videos. Generally, VSI-Bench structures tasks into two formats: Multiple-Choice Answer and Numerical Answer. These formats span eight specific capabilities categorized into three domains: configurational reasoning (including object counting, relative/absolute direction, and route planning); measurement estimation (object/room size and absolute distance); and spatiotemporal reasoning (appearance order). In addition to VSI-Bench, we extend our evaluation to the Static Understanding subset of **STI-Bench** (Li et al., 2025e) to assess the model's generalization on precise geometric perception. While STI-Bench covers a broad spectrum of video tasks, we strictly focus on its static components, specifically Dimensional Measurement, Spatial Relation, and 3D Video Grounding. These tasks demand precise inference of scene properties independent of temporal dynamics.

**Benchmark Models.** To demonstrate the generality and effectiveness of **ReRe** across diverse architectures, we instantiate our framework upon four state-of-the-art open-source MLLM families: Qwen2.5-VL (Bai et al., 2025b), Qwen3-VL (Bai et al., 2025a), InternVL2.5 (Chen et al., 2024b), and InternVL3 (Zhu et al., 2025). We select these models as our foundational backbones due to their superior performance on standard video understanding benchmarks. Furthermore, to position our results within the broader capabil-

*Table 2.* **Performance on STI-BENCH (Static Subset).**

| Methods | Avg. | Dim. Meas. | Spatial Rel. | 3D Video Grounding |
|---|---|---|---|---|
| *Proprietary Models (API)* | | | | |
| GPT-4o | 31.0 | 24.9 | 49.6 | 28.1 |
| Claude-3.7-Sonnet | 37.0 | 31.8 | 49.0 | 36.3 |
| Gemini-2.0-Flash | 36.9 | 33.7 | 50.0 | 33.7 |
| Gemini-2.5-Pro | 37.1 | 34.2 | 53.4 | 32.3 |
| QWEN3-VL-2B | 22.2 | 18.0 | 31.5 | 21.8 |
| **OURS** | $30.2_{+8.0\%}$ | $24.6_{+6.6\%}$ | $50.0_{+18.5\%}$ | $26.2_{+4.4\%}$ |
| QWEN3-VL-4B | 29.7 | 29.8 | 41.8 | 24.3 |
| **OURS** | $34.4_{+4.7\%}$ | $33.6_{+3.8\%}$ | $48.6_{+6.8\%}$ | $28.7_{+4.4\%}$ |
| QWEN3-VL-8B | 27.9 | 29.4 | 43.8 | 19.2 |
| **OURS** | $30.9_{+3.0\%}$ | $29.1_{-0.3\%}$ | $44.5_{+0.7\%}$ | $26.2_{+6.9\%}$ |
| INTERNVL2.5-4B | 30.5 | 25.9 | 43.1 | 29.0 |
| **OURS** | $30.6_{+0.1\%}$ | $22.2_{-3.7\%}$ | $43.2_{+0.1\%}$ | $32.5_{+3.5\%}$ |
| INTERNVL2.5-8B | 32.1 | 30.1 | 45.9 | 27.8 |
| **OURS** | $34.8_{+2.7\%}$ | $33.2_{+3.1\%}$ | $49.3_{+3.4\%}$ | $29.7_{+1.9\%}$ |
| INTERNVL3-2B | 22.6 | 20.1 | 28.8 | 22.1 |
| **OURS** | $26.7_{+4.1\%}$ | $22.5_{+2.4\%}$ | $43.8_{+15.1\%}$ | $22.7_{+0.6\%}$ |
| INTERNVL3-8B | 24.5 | 20.1 | 44.5 | 19.2 |
| **OURS** | $27.8_{+3.3\%}$ | $28.0_{+8.0\%}$ | $37.0_{-7.5\%}$ | $23.3_{+4.1\%}$ |

ity landscape, we include proprietary models (*e.g.*, Gemini-1.5 (Team et al., 2024) and GPT-4o (Hurst et al., 2024)), following previous work (Yang et al., 2025a).

**Inference Setup.** Our inference process involves specific visual data preparation strategies. For the first phase, we utilize 8 uniformly sampled frames from the original scene video. In contrast, for the second phase, we feed the 1 fps-sampled egocentric video into VGGT (Wang et al., 2025b) for 3D reconstruction, render an allocentric video along the planned trajectory, and uniformly sample 8 frames from it to form $V_{exo}$ (matching the 8-frame budget of $V_{ego}$). Regarding model generation, we adopt a low-temperature setting ($T = 0.1$, top-$p = 0.001$), following previous works (Feng et al., 2025; Wu et al., 2025).

### 4.2. Main Results

**VSI-Bench.** As shown in Table 1, **ReRe** substantially boosts performance across diverse open-source architectures, yielding strong gains such as a 5.8% increase for Qwen3-VL-4B. Notably, this inference-time refinement enables the open-source Qwen3-VL-4B to rival the proprietary GPT-4o, effectively closing the gap between open-weight and commercial SOTA models. Fine-grained analysis reveals that these improvements are primarily driven by enhanced configurational reasoning and measurement estimation, confirming that our cross-view verification mechanism successfully mitigates the spatial hallucinations and occlusion issues inherent in single-turn egocentric perception. This benefit also extends to spatial-reasoning-specialized models. Under our evaluation protocol, **ReRe** improves SpaceR-3B (Ouyang et al., 2025) from 34.69 to 35.96 and SpatialLadder-3B (Li et al., 2025b) from 44.84 to 45.58 on the VSI-Bench average. Since **ReRe** keeps the backbone frozen, these gains require no additional training or data curation, suggesting that **ReRe** as an inference-time framework complements

*Table 3.* **Effectiveness of Revisiting Protocol.** Our sequential revisiting protocol outperforms both joint-reason baselines, *Concat* and *Interleaved*, confirming that the revisiting *process*, rather than mere information availability, is the key driver of performance.

| # | Reasoning Protocol | Avg. | Obj. Count | Abs. Dist. | Obj. Size | Room Size | Rel. Dist. | Rel. Dir. | Route Plan | Appr. Order |
|---|---|---|---|---|---|---|---|---|---|---|
| | | | **Numerical Answer** | | | | **Multiple-Choice Answer** | | | |
| 0 | Baseline | 24.8 | 11.2 | 12.2 | 22.8 | 29.9 | 33.8 | 36.0 | 32.0 | 24.9 |
| 1 | Concat | 25.4 | 10.3 | 13.9 | 25.6 | 30.8 | 33.8 | 36.8 | 32.0 | 21.8 |
| 2 | Interleaved | 25.6 | 6.3 | 17.5 | 31.1 | 30.1 | 30.1 | 39.8 | 27.8 | 15.5 |
| 3 | **ReRe** | **29.5** | **18.0** | **18.8** | **40.0** | **31.1** | **34.1** | 35.3 | **32.5** | 22.0 |

*Table 4.* **Deconstructing the Re-Reasoning Components.** We analyze the necessity of both the revisiting process ("thinking twice") and the complementary view source. $V_{ego}$ denotes the original egocentric video, and $V_{exo}$ denotes the synthesized allocentric video.

| # | PHASE I $V_{ego}$ | PHASE I $V_{exo}$ | PHASE II $V_{ego}$ | PHASE II $V_{exo}$ | Avg. | Obj. Count | Abs. Dist. | Obj. Size | Room Size | Rel. Dist. | Rel. Dir. | Route Plan | Appr. Order |
|---|---|---|---|---|---|---|---|---|---|---|---|---|---|
| | | | | | | **Numerical Answer** | | | | **Multiple-Choice Answer** | | | |
| 1 | ✓ | | | | 24.8 | 11.2 | 12.2 | 22.8 | 29.9 | 33.8 | 36.0 | 32.0 | 24.9 |
| 2 | | ✓ | | | 27.3 | 16.8 | 17.9 | 39.2 | 23.6 | 27.5 | 34.7 | 30.4 | 20.1 |
| 3 | ✓ | | ✓ | | 23.5 | 12.8 | 13.4 | 31.7 | 15.0 | 29.9 | 28.0 | 32.0 | 21.2 |
| 4 | ✓ | | | ✓ | **29.5** | **18.0** | **18.8** | **40.0** | **31.1** | **34.1** | 35.3 | **32.5** | **22.0** |

*Table 5.* **Impact of Allocentric Trajectories.** *Mid-level Traverse* fails to resolve occlusions and *Bird's-eye Orbit* suffers a viewpoint domain shift, while our *Oblique Sweep* strikes a balance between exposing hidden layouts and preserving canonical viewpoints.

| # | Trajectory | Avg. | Obj. Count | Abs. Dist. | Obj. Size | Room Size | Rel. Dist. | Rel. Dir. | Route Plan | Appr. Order |
|---|---|---|---|---|---|---|---|---|---|---|
| | | | **Numerical Answer** | | | | **Multiple-Choice Answer** | | | |
| 0 | Baseline | 24.8 | 11.2 | 12.2 | 22.8 | 29.9 | 33.8 | 36.0 | 32.0 | 24.9 |
| 1 | Mid-level Traverse | 25.6 | 11.9 | 13.5 | 28.5 | 30.5 | 32.8 | 34.5 | 30.4 | 24.2 |
| 2 | Bird's-eye Orbit | 27.4 | 14.4 | 18.7 | 40.9 | 22.1 | 28.9 | 31.1 | 29.9 | 24.4 |
| 3 | Oblique Sweep (Ours) | **29.5** | **18.0** | **18.8** | 40.0 | **31.1** | **34.1** | 35.3 | **32.5** | 22.0 |

model-side training by improving what the model observes rather than competing with it.

**STI-Bench.** Extending our evaluation to the Static Understanding subset of STI-Bench, **ReRe** demonstrates robust generalization capabilities. As detailed in Table 2, our approach yields consistent performance gains across various model scales. Most notably, it boosts the lightweight Qwen3-VL-2B by a remarkable 8.0 on average, achieving a massive 18.5 gain in Spatial Relation reasoning. Furthermore, InternVL2.5-8B attains an average score of 34.8, effectively surpassing the proprietary GPT-4o (31.0). These improvements validate that our geometry-driven verification mechanism successfully enhances precise spatial reasoning.

### 4.3. Ablations and Analysis

**Effectiveness of Revisiting Protocol.** Using Qwen2.5-VL-7B on VSI-Bench by default, Table 3 compares **ReRe** against three baselines: single-turn inference on the original video (*Baseline*), concatenating two views into one video (*Concat*), and interleaving videos with separate prompts (*Interleaved*). Results indicate that naive view combination is ineffective. "*Concat*" barely improves over "*Baseline*" likely due to disrupted temporal coherence, while "*Interleaved*" yields only marginal gains because the model still lacks a focused verification objective. In contrast, our structured **ReRe** approach achieves a clear performance boost. This confirms that the revisiting process, rather than mere information availability, is the key driver of performance.

**Deconstructing the Re-Reasoning Components.** While

Table 3 confirms the superiority of our revisiting protocol, Table 4 further disentangles the contributions of the view source and the reasoning depth. We investigate two critical questions: *(1) Is "thinking twice" sufficient?* Simply revisiting the *original* video $V_{ego}$ results in performance degradation, falling even below the baseline. This negative result suggests that without new information, re-reasoning may amplify initial hallucinations. It confirms that the main bottleneck is partial observability rather than insufficient deliberation: our improvement stems from *evidence-seeking* cross-view verification, not from merely increased computational depth on unchanged observations. *(2) Is the allocentric view sufficient?* Relying solely on the synthesized view $V_{exo}$ also underperforms the full **ReRe** pipeline. While it captures geometry, it lacks the fine-grained visual details of the original footage. This validates our design choice to synergize *egocentric semantics* ($V_{ego}$) with *allocentric structural disambiguation* ($V_{exo}$).

**Impact of Allocentric Trajectories.** Table 5 compares our *Oblique Sweep* against two alternative trajectory designs: the horizontal *Mid-level Traverse* and the top-down *Bird's-eye Orbit*. All three trajectories share the same Geometry-to-Video pipeline and differ only in camera extrinsics, so their rendering cost is identical. Results show that trajectory design is critical for effective verification. The *Mid-level Traverse* offers minimal improvement over the baseline, as its horizontal perspective fails to resolve the occlusions inherent in the original view. Conversely, the *Bird's-eye Orbit*, while providing a comprehensive overview, suffers from a viewpoint distribution mismatch. Its strictly vertical

*Table 6.* **Sample-Level Flip Analysis**. Among samples changed by **ReRe**, positive flips outnumber negatives by a 2.52:1 ratio, confirming that corrections substantially outweigh errors.

| Flip Direction | % of Changed Samples | % of All Samples |
|---|---|---|
| Positive: Baseline wrong → **ReRe** correct | **71.6%** | **11.21%** |
| Negative: Baseline correct → **ReRe** wrong | 28.4% | 4.44% |

*Table 7.* **Inference Efficiency of ReRe** on a single A100 GPU. VGGT dominates the latency while rendering is negligible.

| Setting | MLLM ($V_{ego}$) | VGGT | Render | MLLM ($V_{exo}$) | Total | VSI Avg | $\Delta$ |
|---|---|---|---|---|---|---|---|
| Single-turn baseline | ∼1 s | – | – | – | ∼1 s | 30.5 | – |
| **ReRe** (100 frames) | ∼1 s | ∼9 s | < 1 s | ∼1 s | ∼11 s | **35.8** | +5.2 |
| **ReRe** (20 frames) | ∼1 s | ∼2 s | < 1 s | ∼1 s | ∼4 s | 33.3 | +2.8 |

viewpoint lies far from typical viewpoints in the MLLM's pre-training distribution, leading to recognition failures. Our *Oblique Sweep* achieves the best performance by striking a strategic balance: its elevated angle exposes hidden spatial layouts, while its oblique perspective preserves sufficient canonical visual features for the MLLM to maintain semantic understanding.

**Sample-level Flip Analysis.** To decompose how **ReRe**'s gains arise, we analyze sample-level answer changes on VSI-Bench using Qwen3-VL-8B. As shown in Table 6, 15.65% of samples are flipped after Re-reason: among these changed samples, positive flips (Baseline wrong → **ReRe** correct) account for 71.6% while negative flips account for only 28.4%, a 2.52:1 ratio (or 11.21% vs. 4.44% of all samples). This dominance reveals the underlying mechanism: although monocular 3D reconstruction inherently introduces noise, MLLMs reliably extract useful coarse structural cues from $V_{exo}$, showing that *the information gain from coarse geometry outweighs its inevitable noise*.

**Inference Efficiency.** We profile the per-sample latency of **ReRe** on a single A100 GPU with Qwen3-VL-8B as the backbone (Table 7). The full pipeline runs end-to-end in ∼11 s per sample. The dominant cost is the VGGT forward pass (∼9 s). Rendering is negligible (< 1 s), and the second MLLM call is comparable to the first since both consume only 8 frames. This decomposition indicates that the additional inference cost of our revisiting protocol is bounded by the 3D backbone rather than the reasoning protocol itself. Because the framework is agnostic to the specific 3D backbone, this cost can be substantially reduced by faster geometry priors. As shown in Table 7, reducing the VGGT input from the default 100 frames to 20 frames cuts total latency to ∼4 s while still retaining a +2.8 gain over the single-turn baseline.

## 5. Limitations and Discussion

**Robustness to Imperfect Geometry.** **ReRe** relies on monocular 3D reconstruction, which is inherently ill-posed and produces imperfect geometry. Rather than requiring metric-perfect reconstruction, we tolerate this through three

design choices: (1) VGGT fuses the full input video into a unified 3D point cloud, so regions occluded in one frame are often recovered from others; (2) low-confidence points are filtered out before rendering (Sec. 3), leaving uncertain areas blank to avoid fabricating spurious content. Frozen MLLMs tend to treat such blank regions as unobserved blind spots rather than physical objects, avoiding false negatives; (3) $V_{ego}$ remains the semantic anchor, with $V_{exo}$ serving only as cross-view verification, so the model may fall back to the original video when the novel view is incomplete. Empirically, the information gain from coarse geometry consistently outweighs its noise: **ReRe** maintains positive gains across diverse backbones and benchmarks (Tables 1, 2 and 6). Since **ReRe** is agnostic to the underlying 3D backbone, future 3D priors with more accurate geometry and confidence estimates should further reduce this noise without changing **ReRe** itself.

**Computational Considerations and Future Acceleration.** **ReRe**'s two-phase design incurs an additional ∼10 s per sample over single-turn inference on an A100 (Table 7), with most of the cost coming from the 3D reconstruction step. While our contribution focuses on the revisitable reasoning paradigm rather than 3D efficiency, **ReRe**'s modular design directly inherits efficiency gains as 3D priors advance. Table 7 already demonstrates this: reducing the VGGT input to 20 frames brings total latency to ∼4 s while preserving most of the gain. Two further directions can reduce this cost: (1) emerging faster VGGT variants such as FastVGGT (Shen et al., 2026) and LiteVGGT (Shu et al., 2025), together with a number of concurrent training-free accelerators leveraging sparse attention or token compression, are natural drop-in replacements; and (2) in real-world deployment, the geometry-to-video pipeline can be precomputed and cached per scene, amortizing the VGGT cost across multiple spatial queries in the same environment.

## 6. Conclusion

In this work, we address the inherent viewpoint limitations of egocentric spatial reasoning by proposing **Reason, then Re-reason** (**ReRe**), a *training-free* framework that reformulates inference as a revisitable process of hypothesis formation and verification. By leveraging a novel *Geometry-to-Video* pipeline to synthesize complementary allocentric views, **ReRe** enables models to explicitly validate their reasoning against 3D structural evidence without requiring architectural modifications. Extensive evaluations on VSI-Bench and STI-Bench validate our approach, demonstrating that treating geometry as observable evidence effectively resolves spatial ambiguities. More broadly, we hope this revisitable, evidence-driven reasoning paradigm extends to diverse multimodal tasks beyond spatial reasoning.

## Acknowledgments

This work is supported by the National Natural Science Foundation of China (No. 62306178), STCSM (No. 22DZ2229005), and 111 plan (No. BP0719010).

## Impact Statement

This paper presents work whose primary goal is to advance the field of Machine Learning, specifically in the domain of video-based spatial reasoning for Multimodal Large Language Models (MLLMs). Our framework, **ReRe**, aims to improve the reliability and accuracy of spatial perception in egocentric scenarios. This advancement has potential positive implications for the development of embodied AI agents, robotics, and assistive technologies that require robust environmental understanding. While we do not foresee immediate negative societal consequences, we acknowledge that improvements in visual surveillance capabilities carry general ethical considerations common to the field of computer vision.

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

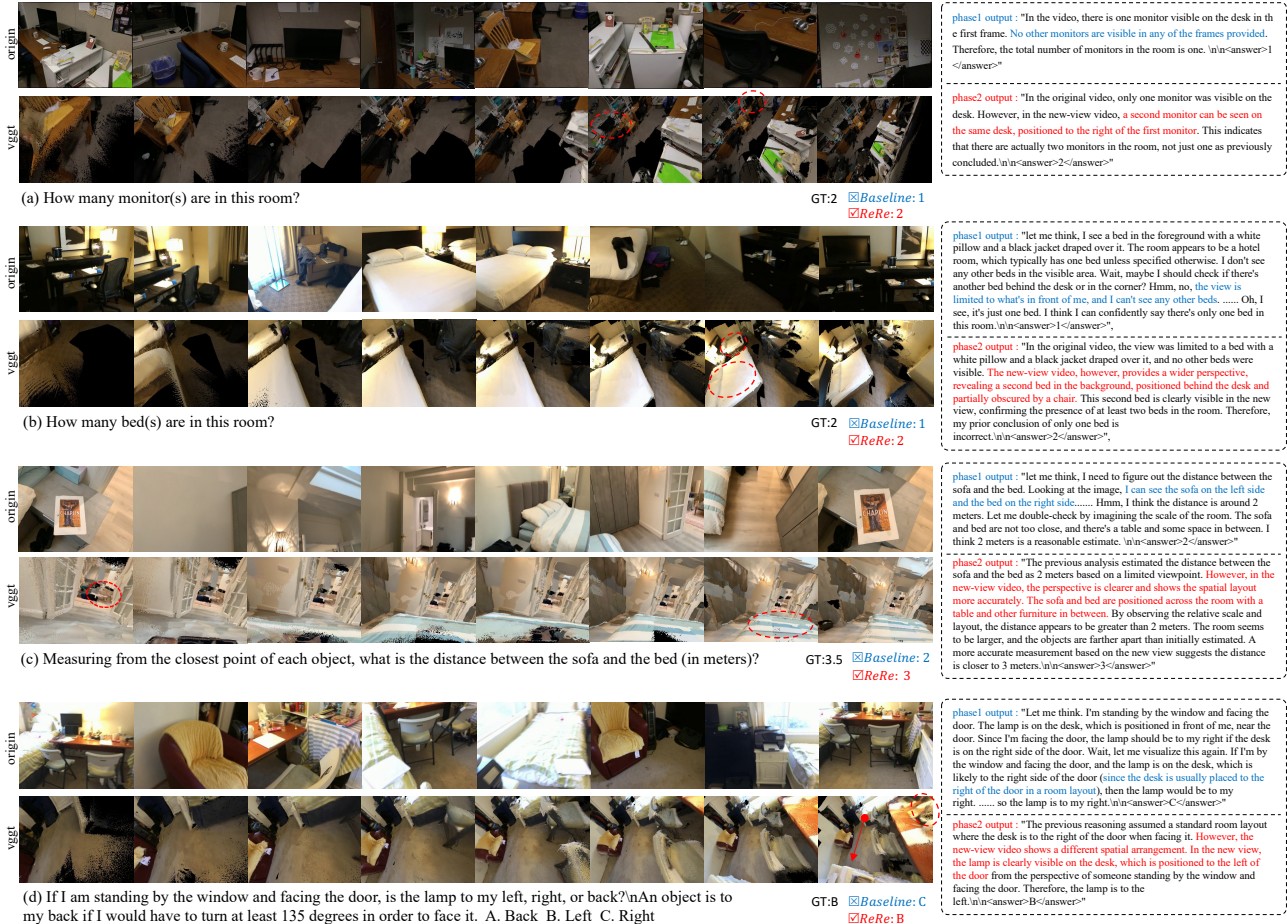

(a) How many monitor(s) are in this room?    GT:2 ☒Baseline: 1 ☑ReRe: 2

(b) How many bed(s) are in this room?    GT:2 ☒Baseline: 1 ☑ReRe: 2

(c) Measuring from the closest point of each object, what is the distance between the sofa and the bed (in meters)?    GT:3.5 ☒Baseline: 2 ☑ReRe: 3

(d) If I am standing by the window and facing the door, is the lamp to my left, right, or back?\nAn object is to my back if I would have to turn at least 135 degrees in order to face it. A. Back B. Left C. Right    GT:B ☒Baseline: C ☑ReRe: B

Figure 5. **Qualitative Results on VSI-Bench.** We visualize how **ReRe** resolves spatial ambiguities in (a)-(b) Object Counting, (c) Absolute Distance, and (d) Relative Direction.

## A. Qualitative Results

In Figure 5, we visualize how our **ReRe** framework corrects erroneous initial judgments by leveraging newly synthesized geometric evidence. We present four representative cases illustrating how the Re-reason Phase resolves spatial ambiguities caused by incomplete egocentric observations. Specifically, for object counting, the synthesized novel views reveal previously unobserved objects: a second monitor on the same desk in (a) and a second bed outside the original visible region in (b), enabling the model to correct its under-counting errors. For absolute distance estimation in (c), the expanded view better exposes the spatial separation between the sofa and the bed, together with the intervening furniture, guiding the model to revise its underestimated distance. Finally, for relative direction reasoning in (d), the synthesized view clarifies the configuration of the window, door, and lamp, allowing the model to replace its incorrect right-side prediction with the correct left-side relation.

## B. Prompt Template

> **Prompt Example for ReRe**
>
> *Reason Phase: Hypothesis Formation*
>
> ```
> {Question}
> Please think about this question as if you were a human pondering deeply. Your
> objectives are as follows:
> ```

```
1. **Observe** the video carefully and describe the key visual elements.
2. **Infer** a plausible answer even if visual information is incomplete.
3. **Conclude** with a final answer.

Engage in an internal dialogue using expressions such as 'let me think', 'wait',
'Hmm', 'oh, I see', 'let's break it down', etc, or other natural language thought
expressions. It's encouraged to include self-reflection or verification in the
reasoning process. Provide your detailed reasoning between the <think> and </think>
tags, and then give your final answer between the <answer> and </answer> tags.
{Answer Format Constraint}
```

*Re-Reason Phase: Cross-View Verification*

```
### Context Injection
Here is your previous reasoning and answer on the original video:{Round 1 Output}
Use this as a baseline for comparison with the new-view video.

### Instruction
{Question}
You previously analyzed the original video and provided an answer. The answer may be
incorrect due to limited viewpoints.
Now you will watch a VGGT-reconstructed new-view video of the same scene.

Your task:
1. **Compare** old vs. new views.
2. **Reflect** on whether your prior conclusion holds. If the question relates to
temporal order, primarily maintain your answer from the first round.
3. **Confirm** your final answer.

Follow this format strictly:
<think>put your Step-by-step reasoning process here</think>
<answer>put your specific answer here</answer>

Rules:
- Do not output text outside tags.
- Only one final answer.
- Avoid vague terms like 'around' or 'approximately'.
{Answer Format Constraint}
Let's think step by step about the comparison.
```

*Answer Format Constraints (Type Template)*

```
[Multiple Choice]
 Please provide only the single option letter (e.g., A, B, C, D, etc.) within the
<answer> </answer> tags.

[Regression]
 Please output exactly one numeric value inside <answer> </answer>. The <answer> tag
must contain only digits and an optional decimal point (e.g., <answer>12.5</answer>).
Do NOT include units, text or ranges inside the <answer> tags.
```

