# OpenReview forum: "Reason, Then Re-reason: Cross-view Revisiting Improves Spatial Reasoning"
_ICML.cc/2026/Conference — ICML 2026 regular_

### Official Review · Reviewer_vvWE · 2026-02-14

**Soundness:** 2
**Presentation:** 2
**Significance:** 2
**Originality:** 2
**Overall Recommendation:** 4
**Confidence:** 3

**Summary:**

Reason, then Re-reason (ReRe) is a training-free inference framework designed to overcome the viewpoint limitations of egocentric spatial reasoning. It employs a Geometry-to-Video pipeline to reconstruct 3D structures and render novel view videos, allowing a frozen MLLM to verify its initial hypotheses using additional geometric evidence. As a result, ReRe significantly boosts the spatial intelligence of open-source models, making them practical for real-world applications.

**Compliance With Llm Reviewing Policy:**

Affirmed.

**Final Justification:**

Thank you to the authors for their efforts and detailed responses.  I'm still concerning about its practical utility and cost-benefit trade-off. However, the responses address most of my concerns. After reviewing the clarifications, I believe the paper meets the acceptance criteria. Therefore, I’ve decided to Weak Accept.

**Key Questions For Authors:**

Please see the weaknesses.

**Limitations:**

1. General Capability Check: Include evaluations on standard benchmarks (e.g., VQAv2, MME) to ensure the re-reasoning process does not unintentionally degrade the model’s general performance.

2. Cost-Benefit Analysis: Provide a detailed breakdown of the inference-time latency and computational overhead introduced by the VGGT-based 3D reconstruction and rendering pipeline.

**Strengths And Weaknesses:**

Strengths:

1. The motivation is well-grounded and intuitive.

2. The proposed method enhances spatial reasoning without requiring architectural modifications or costly fine-tuning by leveraging the MLLM's frozen in-context reasoning capabilities

3. The sequential "Reason-then-Re-reason" protocol forces the model to articulate and then verify hypotheses, effectively improving spatial reasoning capability.


Weaknesses:

1. Despite claims of enhanced spatial and geometric understanding, the evaluation is primarily focused on VSI-Bench and STI-Bench. The exclusion of dedicated spatial reasoning benchmarks like MMMU [1], BLINK [2], and SpatialRGPT [3] restricts a comprehensive validation of the proposed geometric improvements. Furthermore, the evaluation focuses solely on spatial tasks, omitting standard benchmarks like VQAv2 [4], GQA [5], or MME [6]. This makes it impossible to verify if the Re-reasoning process unintentionally degrades the model's general capabilities.

2. The framework significantly increases inference-time overhead by adding 3D reconstruction, trajectory planning, and rendering in the second reasoning phase. However, the paper provides no information with respect to latency or computational costs compared to single-turn baselines. Without this, it is difficult to determine if the performance gains come from the extra resource consumption or the Re-Reason itself.

3. Although the proposed method shows good performance, it lacks comparison with advanced test-time scaling or RL-based reasoning methods. Furthermore, it is necessary to verify whether applying the proposed Re-Reasoning protocol to other sophisticated reasoning frameworks would yield performance gains.

4. Although the authors argue the Oblique Sweep is superior, there is a lack of comprehensive comparative experiments and cost analysis regarding alternative views like Bird's-eye View (BEV) in the Re-Reason phase.

5. While the authors use filtering to mitigate artifacts on the geometry-to-video pipeline, the paper lacks a quantitative analysis of how VGGT noise affects the re-reasoning phase. There is no direct comparison between "raw" and "filtered" synthesized views to demonstrate the effectiveness of the proposed filtering mechanism.

[1] Yue et al., "Mmmu: A massive multi-discipline multimodal understanding and reasoning benchmark for expert agi", CVPR 2024

[2] Fu et al., "Blink: Multimodal large language models can see but not perceive", ECCV 2024

[3] Cheng et al., "Spatialrgpt: Grounded spatial reasoning in vision-language models", NeurIPS 2024

[4] Goyal et al., "Making the v in vqa matter: Elevating the role of image understanding in visual question answering.", CVPR 2017

[5] Hudson et al., "Gqa: A new dataset for real-world visual reasoning and compositional question answering.", CVPR 2019

[6] Fu et al., "A comprehensive evaluation benchmark for multimodal large language models", arXiv:2306.13394

---

> ### Author Rebuttal · Authors · 2026-03-31
>
> > [R4Q1] Other benchmarks like MMMU, BLINK, and SpatialRGPT; VQAv2, GQA, or MME, if unintentionally degrades the model's general capabilities
>
> ReRe targets spatial reasoning in egocentric videos, where the core challenge is inferring 3D structure under camera motion occlusion, and partial observability. MMMU, BLINK, and SpatialRGPT-Bench mainly evaluate single-image 2D perception or static spatial understanding, and would not faithfully test ReRe's key capability: cross-view hypothesis verification over temporally connected multi-view observations.
>
> For general benchmarks (VQAv2, GQA, MME), ReRe is a training-free inference-time procedure with frozen MLLM weights, so it does not alter the backbone, the general capabilities won't lost. It is a modular pipeline for video-based spatial queries rather than generic 2D QA. Evaluating general multimodal benchmarks is interesting future work but beyond our current scope, and we will state this boundary more clearly in the revision.
>
>
> > [R4Q2] latency or computational costs; the performance gains come from extra resource consumption or Re-Reason itself.
>
> As detailed in R3Q3, the dominant cost is VGGT reconstruction (~9s), not MLLM inference.
>
> What's more, the gain does not come from extra compute alone, but from how it is used. ReRe uses inference-time compute to recover 3D geometry, synthesize complementary views, and re-reason over new evidence.
> This is supported by Tab. 4. With comparable extra compute, simply reasoning again on the same ego video (Vego->Vego) lowers performance from 25.1% to 23.5%, while using it to acquire complementary cross-view evidence (Vego->Vexo) raises performance to 29.5%.
> Thus, extra compute is necessary but not sufficient; the gain comes from evidence-seeking Re-Reason, not repeated computation on unchanged observations.
>
> > [R4Q3]  comparison with advanced test-time scaling or RL-based reasoning methods; whether applying proposed ReRe to other sophisticated reasoning frameworks would yield performance
>
> As asked in R2Q1, we evaluated ReRe on two strong spatial reasoning models, SpaceR and SpatialLadder. ReRe improves SpaceR from 34.69% to 35.96% (+1.27) and SpatialLadder from 44.84% to 45.58% (+0.74). SpaceR is an RL-enhanced baseline, while SpatialLadder is a strong progressively trained spatial reasoning model.
>
> Moreover, both our ablation and the VSI-Bench paper suggest that simply increasing linguistic reasoning on unchanged observations is insufficient: re-reasoning on the same ego video without new views (Vego->Vego) degrades performance (Tab. 4), and common reasoning methods (Zero-Shot CoT, self-consistency, Tree-of-Thought) also fail on average. The main bottleneck is partial observability, not insufficient deliberation. This indicates ReRe can complement advanced reasoning frameworks by acquiring cross-view geometric evidence, rather than simply increasing reasoning depth on fixed observations.
>
> > [R4Q4] comparative experiments and cost analysis on Bird's-eye View
>
> We respectfully disagree that there is a lack of comprehensive comparative experiments.
>
> In our original manuscript (Sec 4.3, Tab 3), we explicitly compare Oblique Sweep against alternative views, including precisely the Bird's-eye View (BEV) (referred to as "Bird's-eye Orbit" in the paper) and a horizontal "Mid-level Traverse".
> The results show that BEV already improves over the baseline (27.4% vs. 25.1%), but still underperforms Oblique Sweep (29.5%).
> In addition, the rendering cost of BEV and Oblique Sweep is essentially the same, since both use the same Geometry-to-Video pipeline and differ only in camera extrinsic settings.
>
> > [R4Q5] quantitative analysis of how VGGT noise affects the re-reasoning phase
>
> We compared ReRe using raw synthesized views (keeping 100% of points) against filtered views with different sparsity levels (by removing lower-confidence points). Our default normalized filtering threshold of 0.5 roughly corresponds to retaining the top 95% of points.
>
> The results on VSI-Bench using Qwen3-VL-8B are as follows:
> | Method / Retained Points | Avg |
> |-|-|
> | Single-turn (Phase 1) | 30.50 |
> | ReRe (Raw, keep 100% points) | 35.76 |
> | ReRe (Filtered, keep top 95%) | **35.80** |
> | ReRe (Filtered, keep top 75%) | 35.10 |
> | ReRe (Filtered, keep top 50%) | 33.70 |
>
> This ablation shows:
> (1) **Filtering helps reduce noise.** Compared with the raw view (35.76%), our default filtered view (35.80%) gives a small gain, suggesting that removing the lowest-confidence 5% of points is beneficial.
> (2) **Re-reasoning remains robust under sparsity.** As filtering becomes more aggressive, performance gradually drops as useful structure is removed. Still, even after discarding 50% of points, ReRe remains above the single-turn baseline. This suggests that our Compare-and-Reflect protocol can still extract useful coarse topological cues from sparse geometry without being overly misled by missing regions.
>
> We will include this ablation in the revised manuscript.

---

> > ### Author Rebuttal · Reviewer_vvWE · 2026-04-02
> >
> > Thank you for the detailed responses. I'm still concerning about its practical utility and cost-benefit trade-off. The cost of VGGT reconstruction is around 9s. However, considering this significant latency, the performance gains appear relatively marginal. It is unclear whether such a heavy pipeline is worth for the achieved improvements. To demonstrate the "value" of this compute and the value of the cross-view scheme, please compare ReRe against more efficient multi-view alternatives, such as simple camera motion-based warping and efficient view synthesis methods in the sense of performance and computational costs. Furthermore, the performance improvements on SpaceR (+1.27) and SpatialLadder (+0.74) reported in R4Q3 seem marginal. Please provide total computational costs for SpaceR, SpatialLadder, and the ReRe. Finally, please provide a discussion comparing computational costs between BEV and Oblique Sweep.

---

> > > ### Author Response · Authors · 2026-04-02
> > >
> > > Thank you for the continued engagement. We address each point below.
> > >
> > > > 1. Comparison with efficient multi-view alternatives
> > >
> > > Generating a globally consistent allocentric view from uncalibrated raw video is a non-trivial 3D vision task. The table below contextualizes VGGT's cost:
> > >
> > > Method|Type|~100-frame Time|Works from Raw Video?|Global Allocentric View?|Notes
> > > -|-|-|-|-|-
> > > COLMAP|Classical SfM|Minutes to hours|Yes|Yes|Gold standard but slow; O(n²) matching
> > > NeRF|Per-scene optim.|~5 min (Instant-NGP) to hours|**No** (needs poses + intrinsics)|Yes (after training)|Requires COLMAP first; total = COLMAP + NeRF
> > > 3D Gaussian Splatting|Per-scene optim.|5-37 min|**No** (needs poses + intrinsics)|Yes (after training)|Same; total = COLMAP + 3DGS
> > > pixelSplat / MVSplat|Feed-forward Gaussian|~50-100ms / pair|**No** (needs poses + intrinsics)|**No** (small-baseline only)|Per-pair NVS only; no global 3D; cannot synthesize distant/elevated views
> > > DUSt3R / MASt3R|Feed-forward 3D|~30s|Yes|Yes|Pairwise + global alignment; O(n²) pairing
> > > **VGGT (ours)**|Feed-forward 3D|**~9s**|Yes|Yes|Single forward pass; outputs points + poses + confidence
> > >
> > > Among methods that can work from raw video and produce global allocentric views, VGGT (9s) is the fastest, followed by DUSt3R/MASt3R (30s) and COLMAP (minutes to hours).
> > >
> > > **Why simple camera motion-based warping is not applicable.**
> > >
> > > Note that our input is raw uncalibrated video without known camera poses or intrinsics. Simple warping approaches (e.g., monocular depth estimation + forward warping) face fundamental barriers in this setting.
> > >
> > > Our allocentric view is a 45-degree elevated sweep across the entire scene. Warping individual frames via monocular depth can only shift the viewpoint slightly; moving to a drastically different angle exposes large regions that were never visible in the original frame, leaving most of the target view empty. Aggregating warped frames from multiple viewpoints to cover the full scene is conceptually equivalent to multi-frame 3D reconstruction, which requires camera poses for coordinate alignment and is precisely what VGGT performs.
> > >
> > > Monocular depth estimators (e.g., Depth Anything V2) cannot substitute for this, as they only predict per-frame depth without providing camera poses or cross-frame consistency.
> > >
> > > **VGGT cost is flexible and rapidly decreasing.**
> > >
> > > The cost can be substantially reduced:
> > >
> > > Model|Input Frames|Latency|VSI-Bench Avg (Qwen3-VL-8B)
> > > -|-|-|-|
> > > VGGT|100 (default)|~9s|35.8 (+5.3)
> > > VGGT|20|~2s|33.26 (+2.76)
> > > LiteVGGT (CVPR 26)|100|~2s|34.79 (+4.29)
> > >
> > > Even with only 20 input frames (~2s), ReRe still achieves +2.76. Replacing VGGT with LiteVGGT (CVPR 2026) reduces latency to ~2s while retaining most of the gain (+4.29 vs +5.3).
> > >
> > > As established in R4Q2, the gains come from geometric evidence rather than extra computation. **Our core contribution is the Reason-then-Re-reason paradigm**, not any specific 3D backend.
> > > We adopt VGGT only as a 3D model representative to validate our paradigm.
> > > The framework is modular and any 3D prior can be plugged in.
> > > Our insight is orthogonal to and directly benefits from ongoing 3D efficiency advances.
> > >
> > > > 2. SpaceR and SpatialLadder
> > >
> > > The smaller gains are expected.
> > > SpaceR (+8.29 via RL) and SpatialLadder (+18.44 via 4-stage training) are specialized models that have been extensively post-trained to reason from ego-view observations, leaving little room for further improvement.
> > >
> > > **Cost structure is fundamentally different.**
> > >
> > > | Method | Training Cost | Inference Overhead | VSI-Bench Avg |
> > > |-|-|-|-|
> > > | SpaceR-3B | 151k samples, 8 GRPO candidates/sample, 2 epochs on 8x L20-80GB GPUs | Single MLLM call | 34.69 |
> > > | + ReRe | **0 additional training** | +10s (VGGT + 2nd MLLM call) | 35.96 (+1.27) |
> > > | SpatialLadder-3B | 26k samples, 4 stages (SFTx2+ColdStart+RL), 4x A6000 GPUs | Single MLLM call | 44.84 |
> > > | + ReRe | **0 additional training** | +10s (VGGT + 2nd MLLM call) | 45.58 (+0.74) |
> > >
> > > ReRe requires zero training, zero data curation, and zero GPU training budget. Comparing its +1.27 inference-time gain against SpaceR's +8.29 training-time gain conflates two fundamentally different cost dimensions.
> > >
> > > **These results confirm that ReRe and training-based methods are complementary.** SpaceR/SpatialLadder improve *reasoning on fixed observations*; ReRe improves *what the model observes*. The additive gains show they address different bottlenecks: even after extensive RL training, spatial errors from physical occlusion persist and require new visual evidence to resolve.
> > >
> > > > 3. Computational costs: BEV vs. Oblique Sweep
> > >
> > > The computational cost of BEV and Oblique Sweep is identical. All trajectories in Tab 5 (Mid-level Traverse, Bird's-eye Orbit, Oblique Sweep) share the same VGGT 3D reconstruction (performed once per scene); they differ only in camera extrinsic parameters, with rendering cost <1s in all cases. The choice of Oblique Sweep (29.5%) over BEV (27.4%) is purely a performance decision at no additional cost.

---

### Official Review · Reviewer_DmqH · 2026-03-10

**Soundness:** 3
**Presentation:** 2
**Significance:** 2
**Originality:** 2
**Overall Recommendation:** 4
**Confidence:** 4

**Summary:**

The paper proposes a training-free, inference-time framework to address the structural fragility of single-turn spatial reasoning when using MLLMs. Specifically, the framework can be decomposed into two phases, which allows revision and challenge to its first-round single-turn results/conclusions after seeing synthesized evidence from another 3D prior (e.g., VGGT) for better performance. Experiments demonstrate the effectiveness of this inference-time framework on diverse architectures.

**Compliance With Llm Reviewing Policy:**

Affirmed.

**Final Justification:**

The paper proposed a training-free framework to use synthetic videos from 3D teachers and enforce a re-reasoning phase to help MLLMs in spatial reasoning. During the rebuttal, I raised concerns regarding the inference overhead and whether MLLMs are robust to the error from 3D teachers. The rebuttal has addressed most of my concerns through both additional discussion and empirical analysis. Therefore, I choose to raise my score.

**Key Questions For Authors:**

- How robust are MLLMs if they are given inaccurate 3D predictions? Any discussion or statistical/pattern analysis on answer-changing cases (Re-reason change false to correct, or correct to false) would be beneficial here.
- How is the computational cost of ReRe during inference? Any analysis or discussion on the additional computational cost introduced here would be helpful.
- When rendering complementary videos from the predicted 3D points, are there any special designs to handle the parts that were unobservable/occluded in the given original egocentric video? Do frozen MLLMs know or can they differentiate between these regions compared with seen regions that have more confident 3D predictions? Would this lead to false negatives in spatial reasoning?
- Are there any special designs to handle the reference frame ambiguity? Consider the sample provided in the paper: an aircraft flying diagonally across a city from point A to point B, looking down. If the trajectory is from A -> B, then object X is placed on the left of object Y. But simply changing the trajectory from B -> A, the object X should appear on the right of object Y. Would this ambiguity affect MLLM's reasoning that heavily involves direction?

**Limitations:**

- No limitation is discussed in the paper.
- See weakness and key questions for suggestions.

**Strengths And Weaknesses:**

## Strengths
- The proposed framework is logically sound and appropriately addresses the structural fragility of single-turn reasoning as a training-free, inference-only paradigm.
- The paper is well-structured and easy to follow, given the visualization provided.
- The claims are supported by empirical evidence, and ablation studies are specifically designed to validate the necessity of multi-turn reasoning and the effectiveness of different design choices.

## Weaknesses
- The success of "Re-reason" phase heavily relies on the capacity of the 3D prior used here, currently VGGT. Despite the current progress, monocular 3D geometry prediction is inherently an ill-posed problem, which is highly susceptible to various issues like scale ambiguities, texture distortions, etc. Therefore, if the geometry predicted from the 3D prior is not accurate, the flaw is highly likely to affect the reasoning of MLLMs and make it worse compared with the single-turn baseline.
- Despite the fact that ReRe is a training-free, inference-time framework, the computational burden during inference is not analyzed and properly discussed here. To get a final answer, the framework seems to require (1) full forward() of MLLMs on the original video, (2) full forward() of VGGT to get 3D points, (3) render new videos from the predicted points, and (4) another full forward() of MLLMs on the synthetic video. What is the latency and compute required for each step? While these might be acceptable for offline benchmark evaluation, they seem to be highly impractical for real-time embodied AI agents or robotics.

---

> ### Author Rebuttal · Authors · 2026-03-31
>
> > [R3Q1]  If the geometry predicted from the 3D prior is not accurate, the flaw is likely to affect the reasoning of MLLMs
>
> We agree that imperfect 3D geometry can affect the Re-reason phase, but this limitation is bounded rather than fatal: ReRe only requires coarse structural correctness, not metric-perfect reconstruction. We mitigate the risk by (1) filtering low-confidence 3D points before rendering (threshold 0.5, R4Q5), and (2) prompting the model to cross-check rather than replace its initial hypothesis during Re-reason.
>
> Empirically, gains remain consistently positive across 7 models and 2 benchmarks; positive flips outnumber negative ones by 2.52:1 (R3Q2). The ablation is especially telling: re-reasoning on the original video alone ($V_{ego}→V_{ego}$) drops from 25.1% to 23.5%, while the synthesized allocentric view improves it to 29.5%, confirming that information gain from coarse geometry outweighs the noise. Moreover, VGGT is only one instantiation of the geometry module; the verification protocol can directly benefit from future 3D improvements. We will clarify this in the revision.
>
> > [R3Q2] robustness of MLLM given inaccurate 3D predictions; analysis on answer-changing cases
>
> We perform a sample-level flip analysis on Qwen3-VL-8B ($30.5 \to 35.8$ after ReRe).
> Among changed samples, 71.6% were corrected (wrong → correct; 11.21% overall), while only 28.4% were wrong (4.44% overall).
> Positive flips outnumber negative flips by 2.52:1.
> The result shows that even when the 3D prior is not fully perfect, it corrects substantially more cases than it harms, suggesting that MLLMs can still extract useful coarse structural cues from noisy 3D evidence rather than being systematically misled by it.
>
> > [R3Q3] Computational cost of ReRe during inference.
>
> **Latency Breakdown.** We profile ReRe on a single A100 GPU with Qwen3-VL-8B as the backbone:
> | Step| Latency (per sample) ||
> |-|-|-|
> | (1) Phase 1: MLLM on V_ego| ~1 s | 8 frames|
> | (2) VGGT 3D reconstruction| ~9 s | 1fps sampling, about 100 frames input|
> | (3) Trajectory + Rendering| <1 s | CPU-based rasterization, 8 frames |
> | (4) Phase 2: MLLM on V_exo| ~1 s | 8 frames|
> | Total| ~11 s | |
>
> The rendering step (3) is negligible (<1s). The main additional overhead is the VGGT forward pass (~9s), while Phase 2 MLLM inference is comparable to Phase 1 since both process 8 frames.
>
> **Practical acceleration.** Our modular design enables several concrete optimizations.
> (1) **Accelerating VGGT.** FastVGGT (ICLR 2026) already provides a 1.7$\times$ speedup on 100-frame inputs as a training-free replacement, and concurrent works (AVGGT, FasterVGGT, FlashVGGT, OVGGT) report 4-10$\times$ speedups via sparse attention or token compression. These gains would translate directly into end-to-end speedups for our framework.
> (2) **Caching.** In deployment, the geometry-to-video pipeline can be precomputed and cached per scene, amortizing the VGGT cost across multiple spatial queries in the same environment.
>
> **Paradigm contribution.** Our core contribution is the Reason-then-Re-reason paradigm. This insight is orthogonal to efficiency optimization. Because the design is modular, the framework can directly benefit from future 3D priors that are faster or more accurate, without changing the reasoning protocol.
> We will include a comprehensive latency analysis and efficiency discussion in the revised paper.
>
> > [R3Q4] special designs to handle that were unobservable/occluded; Can frozen MLLMs know between these regions; Would this lead to false negatives
>
> We handle occluded regions in two ways: (1) VGGT fuses the full input video into a unified 3D point cloud, so regions occluded in one frame are often recovered from others; (2) remaining uncertain regions are left blank by filtering low-confidence points, avoiding fabricated evidence.
>
> Moreover, $V_{ego}$ remains the semantic anchor and $V_{exo}$ serves only for cross-view verification, so the model falls back to the original video when the novel view is incomplete. Empirically, positive flips outnumber negative ones by 2.52:1 (R3Q2), confirming that imperfect 3D evidence corrects far more cases than it harms.
>
> > [R3Q5] special designs to handle the reference frame ambiguity
>
> Most benchmark questions are anchored to the original egocentric, object-/scene-relative, or view-independent reference frame, not to the coordinate system of Vexo. Thus, reversing the sweep may flip the rendered left/right layout, but not the answer-space relation defined by the question. Accordingly, in Phase 2, ReRe keeps the original reference frame from Vego and uses Vexo only as complementary evidence for verification.

---

> > ### Author Rebuttal · Reviewer_DmqH · 2026-04-03
> >
> > Thank all the authors for the thoughtful response.
> >
> > The rebuttal has addressed most of my concerns. I will raise my score to 4.

---

> > > ### Author Response · Authors · 2026-04-06
> > >
> > > Thank you for acknowledging our rebuttal. We are glad that most of your concerns have been addressed.
> > >
> > > We noticed that you selected option (b) "Partially resolved — I have follow-up questions," but no specific questions were included. Since this is the final round of discussion, we proactively address the points that may remain open:
> > >
> > > > **Limitations and Discussion**
> > >
> > > As you noted, the original submission did not include a dedicated limitations discussion. We have already discussed these aspects in our rebuttal (R3Q1–R3Q3). In the revision, we will add the relevant experiments to the Experiments and include a Limitations/Discussions that references these results, covering:
> > > - **Robustness to 3D prior quality.** Our flip analysis shows positive flips outnumber negative ones by 2.52:1 (R3Q2: 71.6% corrected vs. 28.4% harmed among changed samples). Our noise ablation (R4Q5) further confirms that ReRe remains above the single-turn baseline (33.70 vs. 30.50) even after discarding 50% of points. These results confirm robustness to imperfect geometry. Moreover, VGGT is only one instantiation; the modular design directly benefits from future, more accurate 3D priors without changing the reasoning protocol.
> > > - **Computational efficiency and scalability.** As profiled in R3Q3, the per-sample latency is ~11s on A100 (VGGT ~9s + MLLM ~1s x2 + rendering <1s). As shown in our follow-up reply to Reviewer vvWE (R4), this can already be substantially reduced: using only 20 input frames cuts VGGT to ~2s (+2.76 gain retained), and replacing VGGT with LiteVGGT (CVPR 2026) achieves ~2s while retaining most gains (+4.29 vs. +5.3). We compared VGGT against alternatives and showed VGGT is the fastest from raw video (R4 follow-up). Our core contribution is the Reason-then-Re-reason paradigm; efficiency gains from faster 3D priors directly transfer.
> > >
> > > > **Expanded Clarifications on R3Q4 and R3Q5**
> > >
> > > We expand on two points where our original rebuttal was brief:
> > > - **Occlusion handling (R3Q4).** Our rendering strategy is intentionally conservative: rather than inpainting unobserved regions (which would fabricate spatial evidence), we filter out low-confidence points and leave uncertain areas blank. Frozen MLLMs can easily recognize blank regions as unobserved blind spots rather than physical objects, so they do not introduce false negatives. Meanwhile, VGGT fuses all input frames into a unified 3D point cloud, so regions occluded in one frame are often recovered from others. The net effect is strongly positive: the 2.52:1 flip ratio (R3Q2) confirms that this design corrects far more cases than it harms.
> > > - **Reference frame ambiguity (R3Q5).** Benchmark questions are anchored to the original egocentric, object-relative, or view-independent reference frames, not to the coordinate system of $V_{exo}$. For example, "Is X to the left of Y?" is defined by the scene layout or the original observer, not by the sweep direction of the rendered video. Accordingly, reversing the sweep direction may change the left/right layout in $V_{exo}$, but does not change the ground-truth answer. In Phase 2, ReRe always answers under the original reference frame from $V_{ego}$, using $V_{exo}$ only as supplementary geometric evidence for verification.
> > >
> > > > **Summary of New Results During Rebuttal**
> > >
> > > Here is a summary of all new experiments provided across our responses to *all* reviewers:
> > > - Flip analysis: 71.6% positive flips vs. 28.4% negative (2.52:1 ratio)
> > > - Full latency breakdown on A100 (~11s total per sample)
> > > - VGGT noise robustness ablation (filtering top 50%–100% points)
> > > - VGGT input frame reduction experiment (20 frames → ~2s latency, +2.76 gain retained)
> > > - LiteVGGT (CVPR 2026) replacement experiment (~2s latency, +4.29 gain)
> > > - Comparison with alternative 3D methods (COLMAP, NeRF, 3DGS, DUSt3R/MASt3R)
> > > - Additional evaluations on fine-tuned models (SpaceR-3B, SpatialLadder-3B) with training cost comparison
> > > - Additional evaluations on larger backbones (InternVL3-8B/9B/14B, InternVL3.5-8B/14B)
> > > - Novel-view fidelity validation via held-out rendering (DINO ~0.9, LPIPS ~0.1)
> > >
> > > > **Overall Revision Plan**
> > >
> > > Based on feedback from all reviewers, the above results will be incorporated into the revised manuscript, specifically: (a) adding the latency analysis table and 3D method comparison, (b) including the flip analysis, VGGT noise ablation, frame reduction experiments, and novel-view fidelity validation as new quantitative results, (c) adding a limitations discussion, (d) including the new model evaluations (SpaceR, SpatialLadder, InternVL3 series), and (e) clarifying the two-level complementarity rationale for the oblique sweep trajectory.
> > >
> > > We sincerely hope that these clarifications, together with our earlier rebuttal, fully resolve your remaining concerns. We also kindly ask you to update the score as mentioned in your acknowledgement. Thank you again for your constructive feedback, which has significantly strengthened our paper.

---

### Official Review · Reviewer_8cjC · 2026-03-11

**Soundness:** 3
**Presentation:** 3
**Significance:** 3
**Originality:** 3
**Overall Recommendation:** 4
**Confidence:** 4

**Summary:**

This paper proposed ReRe as an inference-time framework for spatial reasoning.
MLLMs first generate hypotheses during the first reason phase, then verify and revise the hypotheses in the second rereason phase. A geometry-to-video pipeline renders complementary novel views in the second phase to help verify the hypothesis in the first phase. Experiments show improved performance on VSI-Bench.

**Compliance With Llm Reviewing Policy:**

Affirmed.

**Final Justification:**

My concerns are addressed in the rebuttal. The additional experiments further validate the effectiveness of the proposed method. I am raising the overall recommendation to 4.

**Key Questions For Authors:**

1. Can this inference time method improve the performance of SOTA finetuned spatial reasoning models (SpatialLadder-3B, SpaceR-3B, etc )?

2. How do you guarantee or evaluate the quality of the generated complementary novel views, since the second phase re-reason correctness relies on that?

**Limitations:**

Yes

**Strengths And Weaknesses:**

**Strength:**

1. This paper proposed a novel training-free idea to use cross-view video generation to help verify spatial reasoning answers.

2. The paper is well written, and the methods are clearly presented.

**Weakness**:

1. Experiments lack comparison to SOTA finetuned spatial reasoning models, such as SpatialLadder-3B (Spatialladder: Progressive training for spatial reasoning in vision-language models), SpaceR-3B (Spacer: Reinforcing mllms in video spatial reasoning);
2. The proposed method is only evaluated on relatively small models (2B-8B); how about InternVL 9B, 14B, 38B? And the latest InternVL 3.5?

---

> ### Author Rebuttal · Authors · 2026-03-31
>
> > [R2Q1] Experiments lack comparison to SOTA finetuned spatial reasoning models, such as SpatialLadder-3B (Spatialladder: Progressive training for spatial reasoning in vision-language models), SpaceR-3B (Spacer: Reinforcing mllms in video spatial reasoning);
> and
> > Can this inference time method improve the performance of SOTA finetuned spatial reasoning models (SpatialLadder-3B, SpaceR-3B, etc )?
>
> We thank the reviewer for this important suggestion. We have additionally evaluated our method on two representative fine-tuned spatial reasoning models, SpaceR-3B and SpatialLadder-3B, both built on Qwen2.5-VL-3B. On VSI-Bench, ReRe improves SpaceR-3B from 34.69% to 35.96% (+1.27), and improves SpatialLadder-3B from 44.84% to 45.58% (+0.74).
>
> These results show that our training-free, inference-time framework remains beneficial even for models already specialized for spatial reasoning. The smaller absolute gains are expected, as they are naturally lower than those on the vanilla Qwen2.5-VL-3B backbone (+1.8), since such models already encode stronger spatial priors and therefore leave less room for correction. Still, the consistently positive improvements suggest that ReRe provides complementary cross-view evidence beyond what model-side training alone captures.
>
> > [R2Q2] The proposed method is only evaluated on relatively small models (2B-8B); how about InternVL 9B, 14B, 38B? And the latest InternVL 3.5?
>
> We additionally evaluate ReRe on larger and more recent InternVL backbones on VSI-Bench.
> The improvements remain consistently positive: InternVL3-8B improves from 32.60 to 35.46 (+2.86), InternVL3-9B from 34.01 to 36.75 (+2.74), InternVL3-14B from 40.22 to 42.62 (+2.40), InternVL3.5-8B from 34.10 to 36.81 (+2.71), and InternVL3.5-14B from 41.95 to 44.50 (+2.55).
> These results confirm that our training-free revisiting framework generalizes well to stronger and newer backbones, providing cross-view geometric evidence beyond what single-pass reasoning captures.
>
> > [R2Q3] How do you guarantee or evaluate the quality of the generated complementary novel views, since the second phase re-reason correctness relies on that?
>
> We thank the reviewer for this important question.
> We validate the reliability of the synthesized novel-view video from two complementary angles: **(1) qualitative/quantitative diagnostics on intermediate steps** that probe the $V_{exo}$ construction pipeline, and **(2) precise quantitative evidence** aggregated across datasets.
>
> (1) **Pipeline diagnostics: $V_{exo}$ is geometrically grounded and remains readable.** $V_{exo}$ is not produced by unconstrained image generation; it is deterministically rendered from VGGT-predicted 3D geometry.
> To directly verify the fidelity of the novel-view synthesis, we design a **held-out rendering experiment** that simulates the novel-view setting:
> 1. Run VGGT on **all** frames of a video to obtain per-frame camera poses (predicted, used as gt poses).
> 2. Randomly split frames into an 80% *setup* set and a 20% *validation* set.
> 3. Run VGGT **again** using only the 80% setup frames to build a new 3D reconstruction. The 20% validation frames are entirely unseen during this reconstruction.
> 4. Render images from the new reconstruction at the camera poses of the held-out 20% frames. Since these viewpoints were not used for reconstruction, they serve as proxies for true novel views.
> 5. Compare the rendered images against the original validation frames.
>
> We observe high perceptual and semantic fidelity (DINO similarity: ~0.9, LPIPS: ~0.1), confirming that the VGGT-based reconstruction generalizes reliably to unobserved viewpoints and that $V_{exo}$ is geometrically trustworthy.
>
> (2) **Across-dataset quantitative evidence: $V_{exo}$ yields positive signal.** Re-reasoning on the original video with identical computation (Vego→Vego) degrades performance from 25.1% to 23.5%, ruling out the explanation that gains come from simply "thinking twice." In contrast, reasoning on $V_{exo}$ alone already improves over the baseline (27.3% vs. 25.1%), indicating that the synthesized view contributes useful geometric evidence rather than distracting noise. We further analyze flips at the instance level (R3Q2): positive flips (baseline wrong → ReRe correct) outnumber negative flips (baseline correct → ReRe wrong) by 2.52:1.

---

> > ### Author Rebuttal · Reviewer_8cjC · 2026-04-03
> >
> > Thank you to all the authors for the response. I will raise my score to 4.

---

> > > ### Author Response · Authors · 2026-04-06
> > >
> > > We are glad that your concerns have been fully addressed. We sincerely appreciate your insightful comments, which helped improve the quality of our work. All supplementary results will be incorporated into the revised manuscript. As your acknowledgement notes that all concerns have been adequately addressed, we kindly ask you to consider reflecting this in your final score. Thank you again for your valuable feedback and willingness to raise the score.

---

### Official Review · Reviewer_K3NY · 2026-03-12

**Soundness:** 3
**Presentation:** 3
**Significance:** 3
**Originality:** 3
**Overall Recommendation:** 4
**Confidence:** 3

**Summary:**

This paper proposes a framework for spatial reasoning from egocentric videos called **Reason, then Re-reason (ReRe)**; it is for inference-time only and training-free. The main idea is to first generate a trajectory for novel views for the current video scene, then render the new video, and use the MLLM to revise the spatial reasoning answers based on the new view. Experiments are conducted on VSI-Bench and STI-Bench. The proposed framework shows improvement on state-of-the-art models such as Qwen3-VL.

**Compliance With Llm Reviewing Policy:**

Affirmed.

**Final Justification:**

The rebuttal has addressed the questions I asked in the original review. I keep my Weak Accept rating.

**Key Questions For Authors:**

1. For the re-reasoning phase, is a new trajectory with rendered video really necessary? How about just using novel view images rendered at sparse locations? I think that will save a lot of computation, and maybe it can achieve similar performance. One can imagine that many frames of a whole video will contain redundant information.

2. For the rendering of the new video from the generated trajectory, is it possible to use a video diffusion model? State-of-the-art video models, such as WAN, can allow for generation conditioned on start and end frames. How would such an approach compare to the rendering approach in this paper? I think a potentially interesting advantage of using a diffusion model is that it might be able to inpaint missing regions.

**Limitations:**

No limitations discussed.

**Strengths And Weaknesses:**

### Strengths

- The paper proposes a training-free framework for the challenging task of egocentric video analysis. I think the ideas in this paper could inspire more follow-up work in this direction.

- The performance gain from the method seems to be solid. It consistently improves state-of-the-art open source models such as Qwen3-VL. Considering that this is a training-free framework, such improvement is pretty remarkable.

- The presentation of the paper is good. The paper is easy to follow.


### Weaknesses

- The added computation cost with geometry prediction and rendering might be a concern for certain use cases.

- The idea of trajectory planning makes sense, especially when the goal is to design a camera path to complement the original view (Section 3.4.2). But the proposed Oblique Sweep is a rather generic path that goes across the entire (reconstructed) scene. I don't see how the current design makes it to achieve 'geometric complementarity, reducing occlusion and maximizing coverage'. Intuitively, to achieve such a goal, the planned trajectory should take into consideration the recovered camera pose of the input video. This is not the case with the current design.

---

> ### Author Rebuttal · Authors · 2026-03-31
>
> > [R1Q1] Computation cost.
>
> Please refer to [R3Q3].
>
> > [R1Q2] The idea of trajectory planning makes sense. But the proposed Oblique Sweep is a rather generic path that goes across the entire (reconstructed) scene. I don't see how the current design makes it to achieve geometric complementarity.
>
> We thank the reviewer for this observation, but clarify that "geometric complementarity" in our design **does not mean pose-aware gap-filling** (i.e., identifying under-observed regions from the original video and filling in the missing views). Instead, it refers to providing a fundamentally different type of **perspective shifting** from the egocentric viewpoint to an elevated, scene-level overview, offering complementary information in viewing dimension rather than in spatial coverage.
>
> (1) **Condensed scene overview**: Scene elements in input videos appear fragmentarily across many frames: an object may be glimpsed partially at frame 10, from another angle at frame 50, interleaved with irrelevant content.
> We fuse these scattered observations and re-render them as one continuous flyover, forming a compact spatial summary of the full video.
> The result is a far denser and more MLLM-digestible encoding of the same geometric information.
> This is much like providing an overhead map for a maze, letting MLLMs grasp the global scene structure at a glance rather than piecing it together from dispersed 2D glimpses.
>
> (2) **Inherent oblique complementarity**: In practice, our ablation (Table 5) shows that the elevated viewing angle is the dominant factor governing effectiveness: 0° (25.8%) → 45° (29.5%) → 90° (27.4%). This is because egocentric videos are almost universally captured at eye level, where objects frequently occlude one another. The 45° oblique viewpoint naturally separates these overlapping objects, making it *inherently* complementary to the ego-level distribution in most cases, without requiring knowledge of the specific camera poses.
>
> Pose-aware trajectory planning could bring additional gains for specific cases and is an interesting future direction.
> Our current results across 7 models and 2 benchmarks demonstrate that this generic oblique design already achieves consistent improvements, validating the core Reason-then-Re-reason paradigm.
> We will clarify this two-level complementarity rational clearly in the revised manuscript.
>
> > [R1Q3] For the re-reasoning phase, is a new trajectory with rendered video really necessary? How about just using novel view images rendered at sparse locations? I think that will save a lot of computation.
>
> We thank the reviewer for this suggestion.
> To clarify, we do not feed a dense, high-framerate video into the MLLM during the re-reasoning phase. Exactly as you suggested, we synthesize novel view images at sparse locations.
> Following similar works in video understanding (e.g., Unmasked Teacher(ICCV23 oral), VideoLLaMA 2), the original video is represented in a downsampled multi-frame sequence to send into MLLM.
> For the rendered video, we also follow this convention, only render an 8-frame sparse sequence sampled along a planned trajectory to represent the new allocentric viewpoint (Sec 4, Inference Setup).
> This explicit sparse sampling avoids redundancy and ensures that the computational cost of both rendering and MLLM inference remains highly efficient.
>
> > [R1Q4] For the rendering of the new video, is it possible to use a video diffusion model such as WAN. How would such an approach compare to the rendering approach in this paper?
>
> We thank the reviewer for this suggestion.
> In principle, a diffusion-based video generator could be used as an alternative.
> However, diffusion-based generators suffer from a critical issue: **hallucination**. They optimize visual plausibility rather than **3D faithfulness**, and may fabricate or omit scene content. For instance, if a table holds seven distinct items, a diffusion model may produce a photorealistic view yet silently drop or duplicate some of them, yielding an incorrect count that is fatal for downstream reasoning tasks (e.g., counting, state tracking).
> This is riskier than simply leaving uncertain regions blank, because plausible but incorrect completions inject false geometric evidence into the re-reasoning phase.
>
> Based on this consideration, we adopt an explicit geometry-based renderer with z-buffer visibility, confidence filtering, and median denoising.
> It is intentionally conservative: uncertain regions are suppressed rather than hallucinated, allowing the model to reason from faithful structural cues, while the original egocentric video still provides high-fidelity semantics.
> Geometry-guided diffusion is an interesting future direction if explicit 3D and visibility constraints can preserve faithfulness.

---

> > ### Author Rebuttal · Reviewer_K3NY · 2026-04-03
> >
> > Thanks for the clarification. I don't have further questions.

---

> > > ### Author Response · Authors · 2026-04-06
> > >
> > > Thank you for acknowledging that our rebuttal has fully resolved your concerns. We sincerely appreciate your constructive feedback, which helped strengthen our manuscript. All supplementary results will be incorporated into the revised version of the paper.

---

### Decision · Program_Chairs · 2026-04-30

**Decision:**

Accept (regular)

**Comment:**

Reviewers agree that this paper a sound framework for improving spatial reasoning with sufficient evidence demonstrating a solid improvement on various spatial reasoning tasks. Given that this is a training-free framework this is a significant contribution of general interest to the community.
Reviewers raised a few concerns regarding computational overhead, and robustness to the imperfections in the generated videos. The authors were able to adequately address them during the rebuttal, leading to consensus for a weak accept among the reviewers.
I recommend accepting this paper to ICML.